# Measurement of the bound-electron *g*-factor difference in coupled ions

Tim Sailer[1✉], Vincent Debierre[1], Zoltán Harman[1], Fabian Heiße[1], Charlotte König[1], Jonathan Morgner[1], Bingsheng Tu[1], Andrey V. Volotka[2,3], Christoph H. Keitel[1], Klaus Blaum[1] & Sven Sturm[1]

Quantum electrodynamics (QED) is one of the most fundamental theories of physics and has been shown to be in excellent agreement with experimental results[1–5]. In particular, measurements of the electron's magnetic moment (or *g* factor) of highly charged ions in Penning traps provide a stringent probe for QED, which allows testing of the standard model in the strongest electromagnetic fields[6]. When studying the differences between isotopes, many common QED contributions cancel owing to the identical electron configuration, making it possible to resolve the intricate effects stemming from the nuclear differences. Experimentally, however, this quickly becomes limited, particularly by the precision of the ion masses or the magnetic field stability[7]. Here we report on a measurement technique that overcomes these limitations by co-trapping two highly charged ions and measuring the difference in their *g* factors directly. We apply a dual Ramsey-type measurement scheme with the ions locked on a common magnetron orbit[8], separated by only a few hundred micrometres, to coherently extract the spin precession frequency difference. We have measured the isotopic shift of the bound-electron *g* factor of the isotopes $^{20}$Ne$^{9+}$ and $^{22}$Ne$^{9+}$ to 0.56-parts-per-trillion ($5.6 \times 10^{-13}$) precision relative to their *g* factors, an improvement of about two orders of magnitude compared with state-of-the-art techniques[7]. This resolves the QED contribution to the nuclear recoil, accurately validates the corresponding theory and offers an alternative approach to set constraints on new physics.

The theory of quantum electrodynamics (QED) describes the interaction of charged particles with other fields and the vacuum surrounding them. State-of-the-art calculations of these effects allow for stringent tests of fundamental physics, the search for physics beyond the standard model or the determination of fundamental constants[1–5]. One quantity that can be used to perform such tests is the magnetic moment of an electron bound to a nucleus, expressed by the Landé or *g* factor in terms of the Bohr magneton. It can be both experimentally accessed and predicted by theory to high precision. In particular, hydrogen-like ions, with only a single electron left, provide a simple bound-state system that allows for testing the standard model in the extremely strong electric field of the nucleus. In this case, the *g* factor of a free electron is modified by the properties of the nucleus, foremost the additional electric field, but also parameters such as the nuclear mass, polarizability and the charge radius have to be considered. However, studying these effects explicitly proves to be difficult, as the QED contributions and their uncertainties are significantly larger than many of the nuclear effects, resulting in limited visibility ('*g*-factor calculation' in Methods).

One idea to overcome this limitation is to compare the *g* factors of similar ions, by studying the isotopic shift. Here the common identical contributions and their uncertainties do not have to be considered, emphasizing the differences owing to the nucleus. In Table 1, the theoretical contributions and uncertainties to the individual *g* factors of $^{20}$Ne$^{9+}$ and $^{22}$Ne$^{9+}$ and their differences are summarized. For the calculated difference $\Delta g = g(^{20}\text{Ne}^{9+}) - g(^{22}\text{Ne}^{9+})$, the QED contribution to the nuclear recoil can be resolved and tested independently from all common QED contributions. This QED recoil effect arises from the quantized size of the momentum exchange between the electron and the nucleus, and requires a fully relativistic evaluation that goes beyond the Furry picture[9] and the usual external-field approximation[10]. Understanding and confirming this contribution is essential for future *g*-factor measurements of heavier ions or when trying to improve on the precision of the fine-structure constant $\alpha$ (ref. [11]). Furthermore, a precise measurement of the isotopic shift allows searching for physics beyond the standard model, by means of looking for a deviation from the calculated effect. In particular, a mixing of a new scalar boson and dark-matter candidate, the relaxion, of unknown mass $m_\Phi$, with the Higgs boson would mediate an interaction between nucleons and electrons. Such a mixing with different coupling strengths $y_e$ and $y_n$ for electrons and nucleons, respectively, could potentially be directly observed in the isotopic shift owing to the different number of neutrons. Specifically, such a measurement would exhibit a strong sensitivity of the *g*-factor difference[12] for heavy bosons, with a specific energy range of 20 MeV to 1 GeV owing to the close proximity of the electron

[1]Max-Planck-Institut für Kernphysik, Heidelberg, Germany. [2]Department of Physics and Engineering, ITMO University, St Petersburg, Russia. [3]Helmholtz–Institut Jena, Jena, Germany. ✉e-mail: tim.sailer@mpi-hd.mpg.de

**Table 1 | Contributions to the g-factor difference of $^{20}$Ne$^{9+}$ and $^{22}$Ne$^{9+}$ and the final experimental result**

| g-factor theory | (×10$^{-9}$) |
|---|---|
| $^{20}$Ne$^{9+}$ | 1,998,767,277.112(117) |
| $^{22}$Ne$^{9+}$ | 1,998,767,263.638(117) |
| **Difference** | |
| FNS | 0.166(11) |
| Recoil, non-QED | 13.2827 |
| Recoil, QED | 0.0435 |
| Recoil, $(\alpha/\pi)(m_e/M)$ | −0.0103 |
| Recoil, $(m_e/M)^2$ | −0.0077 |
| Nuclear polarization | 0.0001(3) |
| **Δg Total theory** | 13.474(11)$_{FNS}$ |
| **Δg Experiment** | 13.475 24(53)$_{stat}$(99)$_{sys}$ |

The dominating uncertainty stems from the FNS. All digits are significant when no uncertainties are given. $m_e$ and $M$ are the electron and nuclear mass, respectively. For the individual contributions, see Extended Data Table 1.

to the nucleus in a highly charged ion (HCI) ('Setting constraints on new physics' in Methods). The relaxion, if found, could potentially provide a solution to the long-standing electroweak hierarchy problem[13]. To explicitly study the isotopic shift with formerly unavailable resolution, we report on the application of a technique developed to measure the difference between the g factors directly. This method depends on coupling two ions as a well controlled ion crystal within the magnetic field of a Penning trap. In this way, the ions are close enough to be subject to the identical fluctuations of this magnetic field, which otherwise pose strong limitations for the achievable precision. We performed such a measurement in the ALPHATRAP setup[6]. This apparatus consists of a Penning trap[14] in a superconducting 4-T magnet, where the trap and all detection electronics are cooled by liquid helium to about 4.2 K. By combining the magnetic field $B$ and a suitable electrostatic potential, ions can be stored almost indefinitely, limited only by the vacuum quality. A trapped ion's motion can be parametrized by splitting the trajectory into three independent harmonic oscillations that are related to the free cyclotron frequency $\nu_c = \frac{q_{ion}}{2\pi m_{ion}}B$, with the ion charge and mass $q_{ion}$ and $m_{ion}$ respectively, via[14]:

$$\nu_c^2 = \nu_+^2 + \nu_z^2 + \nu_-^2. \qquad (1)$$

For this measurement on $^{20}$Ne$^{9+}$ and $^{22}$Ne$^{9+}$, the modified cyclotron frequencies $\nu_+$ amount to roughly 27 MHz and 25 MHz, the axial frequencies (parallel to the magnetic field) $\nu_z$ to about 650 KHz and 620 KHz, and both magnetron frequencies $\nu_-$ to 8 kHz, respectively. These frequencies can be measured non-destructively through the image currents induced by the oscillating charged particle[15,16]. In addition, the presence of the magnetic field results in an energy splitting $\Delta E = h\nu_L$ of the $m_s = \pm 1/2$ electronic spin states with the Larmor frequency $\nu_L = \frac{geB}{4\pi m_e}$ amounting to about 112 GHz, with the electron charge and mass $e$ and $m_e$, respectively ($h$ is Planck's constant). The orientation $m_s$ of the spin with respect to the magnetic field can be determined by means of the continuous Stern–Gerlach effect[17] in the dedicated analysis trap (AT) (Fig. 1). Here, in addition to the homogeneous magnetic field $B_0$, a quadratic magnetic field gradient or magnetic bottle $B(z) = B_0 + B_1 z + B_2 z^2$ with $B_2 \approx 45$ kT m$^{-2}$ is produced by a ferromagnetic ring electrode. This exerts an additional spin-dependent force on the ion that results in an instantaneous shift of the axial frequency when a millimetre-wave (photon around $\nu_L$ is absorbed. As this magnetic bottle hinders precise frequency measurements, the spectroscopy is performed in the homogeneous magnetic field[6] of the precision trap (PT), where also the cyclotron frequency can be measured

simultaneously to the millimetre-wave excitation. The AT is then solely used for the detection of the spin state and the separation of the ions. The g factor can be extracted from the frequencies[3,7,18]

$$g = 2\frac{\nu_L}{\nu_c}\frac{m_e}{m_{ion}}\frac{q_{ion}}{e}. \qquad (2)$$

Consequently, the independently measured ion masses, as well as the electron mass, pose direct limits on the achievable precision of absolute g-factor measurements. In addition, the inherent magnetic-field fluctuations render it impossible to determine the Larmor frequency coherently on the timescales required to accurately measure the cyclotron frequency. This limits such measurements statistically to low 10$^{-11}$ relative precision even with several months of measurement time, and renders an investigation of the small nuclear effects impractical.

## Coupled ions

To overcome these limitations, we have developed a measurement technique based on the principle of the two-ion balance[8,19]. Here the ions are first prepared separately in the AT to a known electron spin orientation and subsequently merged by placing them in the same potential well of the PT (this process takes about 10 min). After cooling the axial motion of the ions individually, they become coupled on a common magnetron orbit owing to the almost identical frequencies of this mode ($\Delta\nu_- \approx 200$ mHz), whereas the axial and modified cyclotron motions remain uncoupled owing to their large frequency discrepancy ($\Delta\nu_z \approx 30$ kHz and $\Delta\nu_+ \approx 2.5$ MHz). The combined motion, as shown in Fig. 1b, can be parametrized as a superposition of a rotation of both ions with a quasi-static separation distance $d_{sep}$ around a common guiding centre and a rotation of this guiding centre around the trap centre on a radius $r_{com}$. The coupling interactions have been mathematically described and used for mass comparison measurements in ref. [8]. Now, we determine the initial values of $d_{sep}$ and $r_{com}$ by measuring the axial frequency shift resulting from the Coulomb interaction of the ions, as well as the individual absolute magnetron radii (merging and determining the initial configuration takes about 10 min). Subsequently, we are able to transfer canonical angular momentum, or effectively mode radius, from the common mode to the separation mode[20] (see 'Mixing and preparing the coupled state' in Methods), as well as directly cool the separation mode by coupling it to the axial mode. In this way, we have full control over all modes as the axial and cyclotron modes of both ions can still be addressed individually. We apply these tools to prepare the ions with a magnetron separation distance $d_{sep} \approx 400$ μm and a comparably small common mode radius $r_{com}$ (see 'Mixing and preparing the coupled state' in Methods; about 20 min). Now, we perform simultaneous Ramsey-type measurements on the electron spins by irradiating a single millimetre-wave π/2 pulse (see 'Rabi frequency measurement' in Methods) for both ions simultaneously. We then wait for the evolution time $\tau_{evol}$, during which both magnetic moments are freely precessing with their individual Larmor frequencies and finally irradiate the second π/2 pulse (this takes about 5 min, including a determination of $\nu_c$). Subsequently, the ions are separated again (see 'Separation of ions' in Methods; duration 10 min). Finally, the cycle is completed by determining and comparing the spin orientation to the initial state for each ion individually in the AT again. This whole process has been fully automated, requiring about 1 h to complete a cycle. In total, we have performed 479 cycles for the main measurement as well as 174 for the systematic uncertainty analysis. Owing to the fast Larmor precession of 112 GHz, the inherent magnetic-field fluctuations lead to decoherence of the applied millimetre-wave drive frequency with respect to the individual spin precessions already after some 10 ms, as also observed in ref. [21]. However, as the ions are spatially close together, the spins stay coherent with respect to each other as they both experience identical fluctuations. For each evolution time $\tau_{evol}$ of the Ramsey

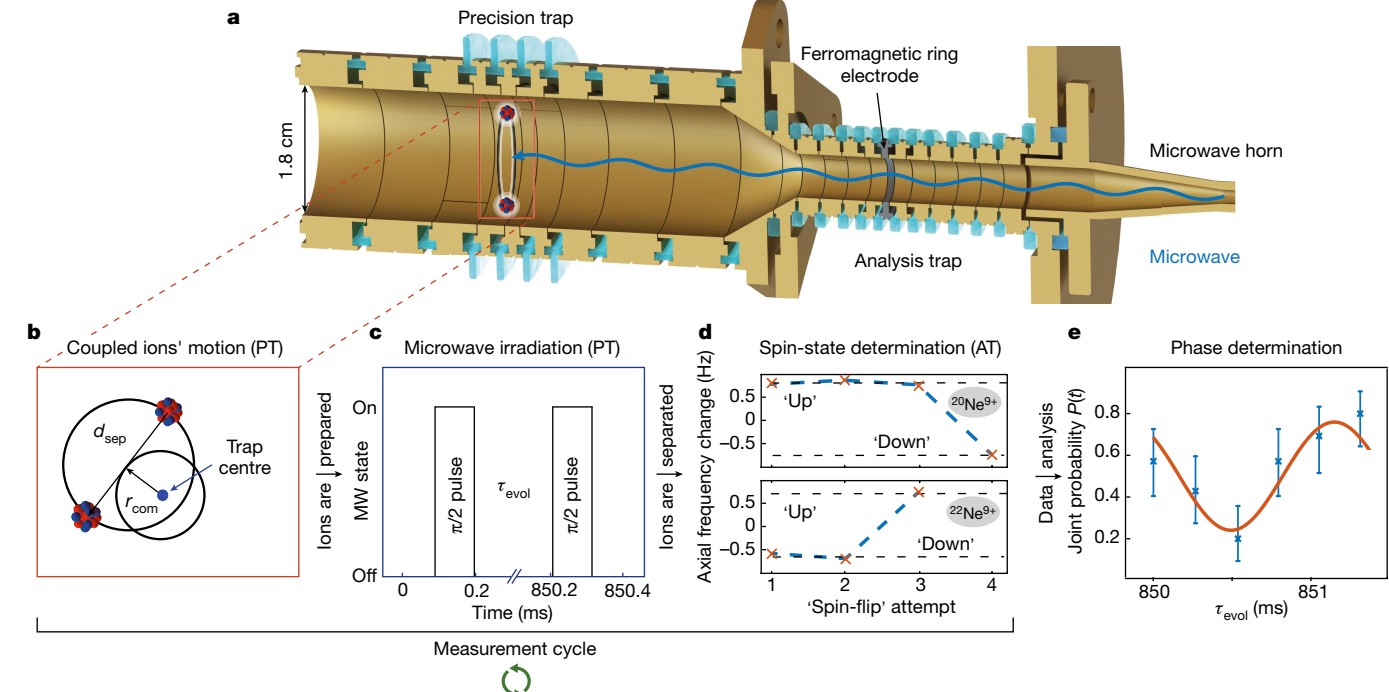

**Fig. 1 | Experimental setup and measurement scheme. a**, The Penning-trap setup, with the coupled ions in the centre of the precision trap. **b**, The ions are prepared on a common magnetron orbit, with a separation distance of $d_{sep} \approx 400\ \mu m$ and a common mode $r_{com} < 100\ \mu m$. The cyclotron radius $r_p$ of each ion is cooled to $r_p \approx 3\ \mu m$ and the axial amplitude to $r_z \approx 18\ \mu m$ when in thermal equilibrium with the resonator circuit at $T = 4.2$ K. **c**, The pulse scheme of the millimetre-wave irradiation. **d**, The change of axial frequency after each attempt to induce a spin transition. Here $^{20}Ne^{9+}$ was found to be in the 'up' state and $^{22}Ne^{9+}$ was found to be in the 'down' state after the measurement sequence, as can be deduced from the observed change. **e**, After several repetitions of such cycles, the coincidental behaviour of the spin-transition rate modulation $P(t)$ is fitted, error bars represent the 68% confidence interval.

scheme, the individual measurement points are distributed over roughly one period of the difference frequency $\Delta \nu_L = \nu_{L1} - \nu_{L2} \approx 758$ Hz. The coherent difference of the precession frequencies can now be extracted from the correlated spin transition probability $P$. Here, the ions behave identically when their individual spins are in phase, or opposite to each other when the spins are out of phase after the evolution time. We can therefore define

$$P = p_{1,SF} \times p_{2,SF} + p_{1,noSF} \times p_{2,noSF},\qquad(3)$$

where $p_{n,SF}$ and $p_{n,noSF}$ are the probabilities for ion $n$ to undergo or not undergo a spin transition, respectively (see 'Fitting function for the Larmor frequency difference' in Methods). This relation encodes the relative phases of the spins to each other but, owing to the loss of coherence with respect to the applied microwave drive, the modulation amplitude $A$ is only ±25%. This joint transition probability is therefore directly modulated by the differential phase of the spins and follows

$$P(t) = A\cos(2\pi(\Delta \nu_L)\,t + \phi_{\tau,0}) + \frac{1}{2},\qquad(4)$$

with an additional phase $\phi_{\tau,0}$ encoding the difference of the Larmor frequencies.

## Data analysis

We have performed measurements for five different sets of evolution times and three different separation distances. Figure 2 shows the modulated probability of a coincidental spin transition occurring for all of these measurements. To extract the Larmor frequency difference, first the total accumulated phase has to be unwrapped. We perform a maximum likelihood fit with a fixed frequency difference, fitting

only the phase $\phi_{\tau,0}$ and amplitude $A$, separately for each evolution time. For all six measurements, the observed amplitude is compatible with a modulation amplitude $A = 25\%$, which confirms the coherent behaviour of the two quantum states for at least up to $\tau_{evol} = 2.2$ s, which is more than a factor 20 longer than the coherence time of the individual spins with respect to an external drive. After unwrapping, a linear fit to those phases measured with the separation distance $d_{sep} = 411(11)\ \mu m$ as well as the calculated initial phase difference (see 'Calculation of the initial phase difference' in Methods) is used to determine the frequency difference and the statistical uncertainty. Systematic shifts are expected to arise owing to the small imbalance of the coupled magnetron motion, which is a consequence of the different ion masses. This causes the ions to experience slightly different magnetic fields and alters their individual Larmor frequencies. The two main contributions are: first, this radial imbalance in combination with a residual $B_2$; and second, a slight shift of the axial equilibrium position caused by a residual deviation from the perfect symmetry of the electrostatic trapping potential, leading to an unequal change of the Larmor frequencies in the presence of a linear axial $B_1$ field gradient. The combined systematic shift has been evaluated (see 'Calculation of the initial phase difference' in Methods) as $6(5) \times 10^{-13}$ relative to the mean Larmor frequency. We specifically stress that our method, although currently experimentally limited by magnetic-field inhomogeneities, could be significantly improved by implementing active compensation coils for $B_1$ and $B_2$ (ref. [22]), possibly extending the precision to the $10^{-15}$ regime. The bottom plot of Fig. 2 shows the residual deviation of each extracted phase with respect to the final frequency difference and uncertainty, corrected for this systematic shift. The grey highlighted data points are for the two measurements performed at a different separation distance, corrected for their expected systematic shift. The agreement of these measurements clearly confirms the systematic correction independently from the

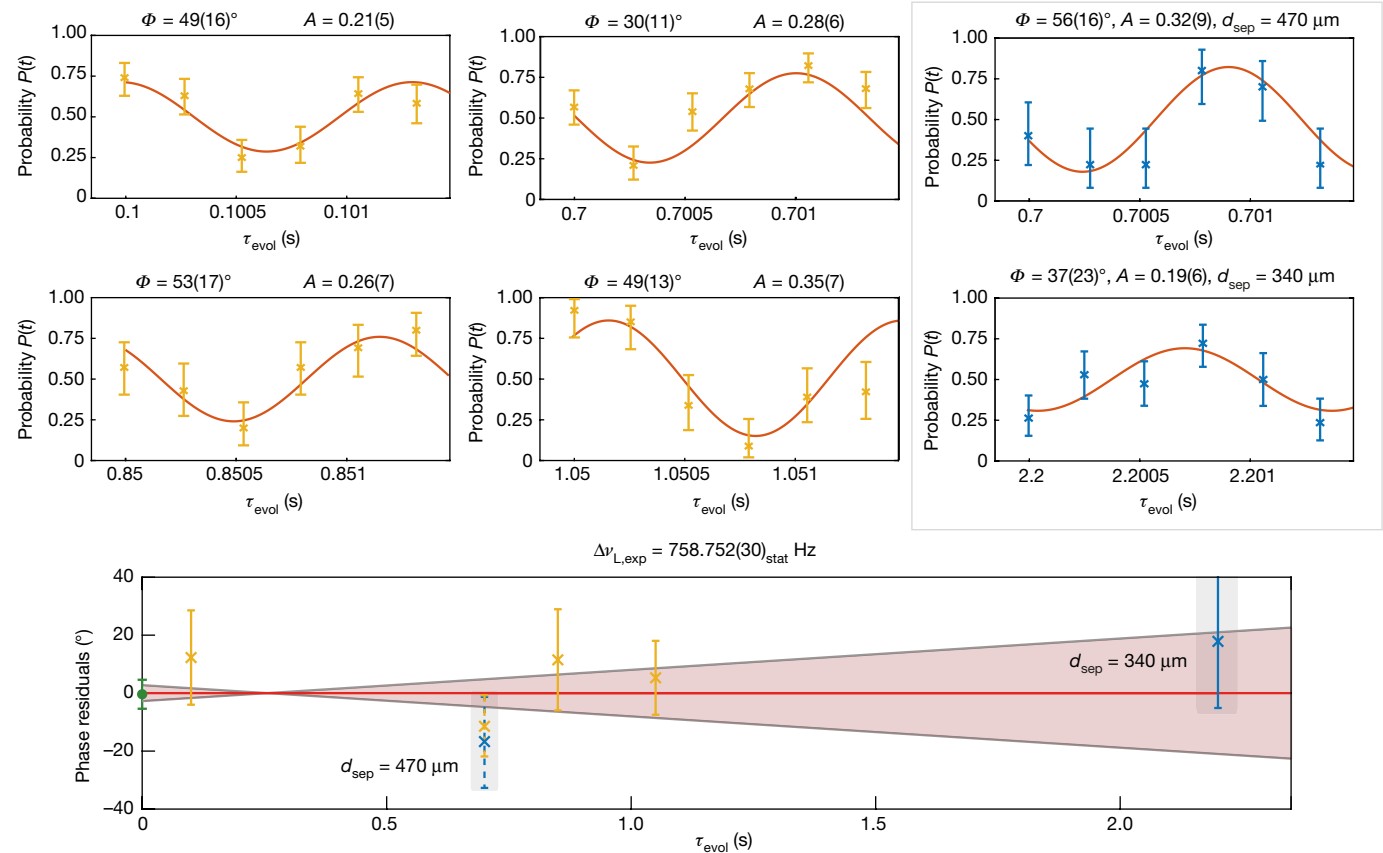

**Fig. 2 | Experimental results.** The top six panels show the individual measurements. The two panels highlighted in grey do not contribute to the statistical uncertainty of the final result and are used only to confirm and correct for systematic effects. The bottom panel shows the residuals with respect to the final frequency, with the $1\sigma$ statistical uncertainty being illustrated in the shaded confidence interval (red). The initial phase (green) stems from numerical calculation. The error bars in the top six panels represent the 68% confidence interval.

calculated correction derived from independent single-ion measurements. The frequency difference of $\Delta\nu_L = 758.752(30)_{stat}(56)_{sys}$ Hz, which corresponds to $\Delta g = 13.475\,24(53)_{exp}(99)_{sys} \times 10^{-9}$ ('Calculation of the $g$-factor difference' in Methods), is in agreement with the theoretical calculation of $\Delta g = 13.474(11)_{FNS} \times 10^{-9}$, limited in precision solely by the uncertainty of the charge-radius difference (finite nuclear size) of the isotopes $\delta\langle r^2\rangle^{1/2} = 0.0530(34)$ fm (ref. [23]). Taking theory as an input instead, our result can thus be applied to improve on the precision of the charge-radius difference by about one order of magnitude $\delta\langle r^2\rangle^{1/2} = 0.0533(4)$ fm. With the agreement between theory and the experimental result, we can set constraints for the scale of the $y_e y_n$ coupling constants, which appear in the new physics search in the Higgs-relaxion mixing scenario ('Setting constraints on new physics' in Methods). Although the bounds obtained here do not improve on already existing bounds (Fig. 3), they are derived using an alternative approach and do not rely on King plot linearity and hence offer a more direct[12] and less ambiguous[24,25] way to search for new physics. The obtained bounds are shown in Fig. 3, along with bounds from other fields of physics (see caption). At the present, our method and H–D spectroscopy are both dependant on charge radii determined by muonic spectroscopy[23,26]. When combining our method with measurements on the $g$ factor of lithium-like ions[27] or the ground-state energy isotope shifts of hydrogen-like ions, improved competitive bounds could be envisaged[28] while additionally gaining independence from assumptions on new physics coupling to muons. The $(g-2)_e n$ bounds are independent from muonic radii, but rely on a combination of multiple neutron-scattering measurements as well as the free-electron $g$ factor[29].

## Conclusions and outlook

We have demonstrated and applied our method to directly measure a $g$-factor difference coherently to high precision. This is a direct test and validation of the hitherto untested QED contribution to the nuclear recoil and paves the way towards further high-precision measurements on heavier ions where this contribution becomes even larger. Furthermore, we are able to improve on the precision of the charge-radius difference by about one order of magnitude using this method, which could be similarly applied to other systems. In addition, we have applied the result of this single isotopic-shift measurement to strengthen the limits on the parameters for the new-physics search via the Higgs-relaxion mixing, showcasing the potential of this approach. Furthermore, this method provides a crucial step towards accessing the weighted difference of $g$ factors[11,30], which has the potential to significantly improve on the precision of the fine-structure constant $\alpha$. Here, the difference between two ions of different nuclear charge $Z$ will have to be measured for both their hydrogen-like (1$s$) and lithium-like (2$s$) states using this method. In addition, a single absolute $g$ factor of low $10^{-11}$ precision is required when choosing ions of the medium $Z$ range, which has already been shown to be experimentally feasible[3]. However, the theoretical calculation of this $g$ factor has to achieve similar precision, which will still require significant work and time. Finally, the possibility to directly compare matter versus antimatter with highly suppressed systematics should be investigated. This method could possibly be applied to directly compare the anti-proton and H⁻ $g$ factors. In this case, the Larmor frequency difference would be mostly defined by the electronic shielding of the H⁻ ion, which would have to

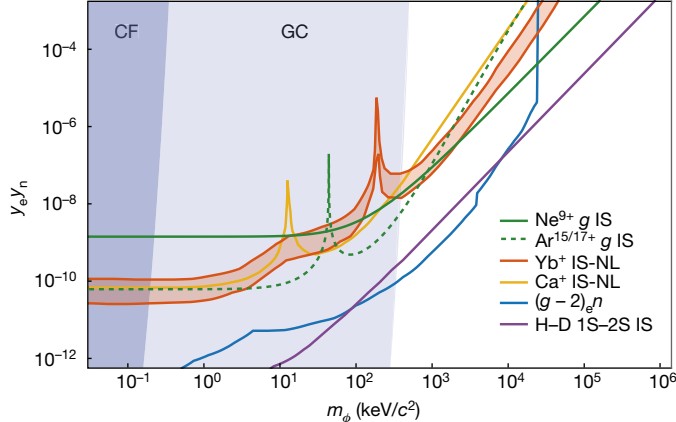

**Fig. 3 | New physics exclusion plot.** Bounds at the 95% confidence level on the NP coupling constant (Methods) as a function of the mass of a new scalar boson. Other shown limits are derived from Casimir force (CF) measurements[33], globular cluster (GC) data[34], isotopic-shift measurements in Ca⁺ (ref. [35]) and in H–D (ref. [36]), and a combination of neutron-scattering data with the free-electron $g$ factor (($g-2$)$_e n$ (ref. [29])). The orange region indicates possible values of the coupling constant derived from isotopic-shift measurements in transition frequencies in Yb⁺ (ref. [37]), under the assumption that the observed King nonlinearity is caused by NP (see also refs. [38,39]). The solid green line shows the limits derived in this work. The dashed green line shows a projected bound from isotopic-shift measurements of the $g$ factor of hydrogen-like and lithium-like argon.

be calculated to similar precision as shown for ³He (ref. [31]). Similar to the mass comparison that was already performed[32], this could enable a direct $g$-factor comparison with significantly reduced systematic effects. If a further comparison of proton and positive anti-hydrogen H̄⁺ becomes experimentally feasible in the future, even the uncertainty of the shielding could be dramatically reduced as well.

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

## Methods

### Mixing and preparing the coupled state

After determining the spin orientations of the individual ions, one ion is excited to a magnetron radius $r_- \approx 600$ µm to prepare for the coupling of the ions. They are now transported into electrodes next to each other, with only a single electrode in between to keep them separated. Subsequently, this electrode is ramped down as quickly as experimentally possible, limited by d.c. filters to a time constant of 6.8 ms to keep any voltage change adiabatic compared with the axial frequencies of several 10 kHz. The potentials are also optimized to introduce as little axial energy as possible during this mixing. Subsequently, both ions are brought into resonance with the tank circuit one at a time by adjusting the voltage to repeatedly cool their axial modes. Once thermalized, the axial frequency is automatically measured and adjusted to the resonance frequency. From the observed shift in axial frequency compared with a single cold ion, the separation distance $d_{sep}$ of the ions can already be inferred, without gaining information about the common mode. At this point, both ions are cooled in their respective cyclotron motions via sideband coupling[15]. The common-mode radius $r_{com}$ of the coupled ions can be measured by applying a $C_4$ field contribution, causing the axial frequency to become dependent on the magnetron radius. With the amplitudes of the axial and reduced cyclotron motion being small, this frequency shift allows for the determination of the root-mean-square (r.m.s.) magnetron radius of each ion. If the common mode is large, the modulation of the magnetron radius, owing to slightly different frequencies of separation and common mode, will lead to visible sidebands owing to the axial frequency modulation.

For small common mode radii, we will simply measure half the separation radius for each ion. In combination with the known separation distance, the common mode radius can now be determined; however, owing to limited resolution of the axial frequency shift and the quadratic dependency, $r_{com} \approx \sqrt{r_{r.m.s.}^2 - \left(\frac{1}{2}d_{sep}\right)^2}$, a conservative uncertainty after the ion preparation of $r_{com} = 0(100)$ µm is assumed. For consistency, we have prepared the ions in the final state and again excited the common mode to a known radius that could be confirmed using this method. In case of a large initial common mode, we first have to cool it. Unfortunately, addressing it directly is complicated, as the separation mode will always be cooled as well. However, using the method described in ref. [20], we are able to transfer the common-mode radius to the separation mode. This requires a non-harmonic trapping field with a sizeable $C_4$, combined with an axial drive during this process. The axial frequency will now be modulated owing to the detuning with $C_4$ in combination with the modulated radius owing to the common mode. As the ion will be excited only when close to the drive, we gain access to a radius-dependant modulation force, which finally allows the coupling of the common and separation modes.

Finally, with the common mode thus sufficiently cooled, we directly address the separation mode, cooling it to the desired value. Owing to the strong axial frequency change during cooling, scaling with $d_{sep}^3$ and typically being in the range of $\Delta\nu \approx 150$ Hz, the final radius cannot be exactly chosen but rather has a distribution that scales with the power of the cooling drive used. Therefore, one can chose to achieve more stable radii at the cost of having to perform more cooling cycles, ultimately increasing the measurement time. We choose a separation distance $d_{sep} = 411(11)$ µm, with the uncertainty being the standard deviation of all measurements as an acceptable trade-off between measurement time and final separation distance distribution. Furthermore, although a smaller separation distance directly corresponds to a decreased systematic uncertainty (Methods), the increased axial frequency shift as well as a deteriorating signal quality of the coupled ions result in a practical limit around $d_{sep} = 300$ µm.

### Measurement sequence

Before irradiating the microwave pulses, the cyclotron frequency is measured via the double-dip technique using $^{22}\mathrm{Ne}^{9+}$. This measurement is required to be accurate to only about 100 mHz, which corresponds to a microwave frequency uncertainty of about 400 Hz, which is neglectable considering a Rabi frequency of over 2 kHz for a spin transition. The microwave pulse is applied at the median of the Larmor frequencies of $^{22}\mathrm{Ne}^{9+}$ and $^{20}\mathrm{Ne}^{9+}$ and therefore detuned from each Larmor frequency by about 380 Hz. This detuning is taken into account when calculating the required time for a π/2 pulse.

### Separation of ions

The strong magnetic bottle, or $B_2$ contribution, that is present in the AT gives rise to a force that is dependent on the magnetic moment of the ion. The main purpose is to allow for spin-flip detection via the continuous Stern–Gerlach effect. In addition, this $B_2$ can be utilized to create different effective potentials for the ions depending on their individual cyclotron radii $r_+$. These give rise to the magnetic moment $\mu_{cyc} = \pi\nu_+ q_{ion} r_+^2$, which then results in an additional axial force in the presence of a $B_2$. To use this effect to separate the coupled ions, one of them is pulsed to $r_+ \approx 800$ µm at the end of the measurement in the PT. Subsequently, both ions are cooled in their magnetron modes, resulting in a state where one ion is in the centre of the trap at thermal radii for all modes while the other is on the large excited cyclotron radius. We verify this state by measuring the radii of both ions to confirm the successful cooling and excitation. Now, we use a modified ion-transport procedure, with the electrode voltages scaled such that the ion with $r_+ > 700$ µm cannot be transported into the AT but rather is reflected by the $B_2$ gradient, whereas the cold ion follows the electrostatic potential of the electrodes. The hot ion is transported back into the PT and can be cooled there, leaving both ions ready to determine their electron spin orientation again, completing a measurement cycle. This separation method has worked flawlessly for over 700 attempts.

### Rabi frequency measurement

To determine the required π/2-pulse duration, a single ion, in this case $^{22}\mathrm{Ne}^{9+}$, is used. We determine the spin orientation in the AT, transport to the PT, irradiate a single microwave pulse and check the spin orientation again in the AT. Depending on the pulse duration $t$, the probability of achieving a change of spin orientation follows a Rabi cycle as

$$P_{SF}(t) = \frac{\Omega_R^2}{\Omega_R'^2}\sin^2(\Omega_R'\pi t),$$
$$\Omega_R' = \sqrt{(\Omega_R^2 + \Delta\Omega_L^2)}. \quad (5)$$

Here, $\Omega_R$ is the Rabi frequency and $\Delta\Omega_L$ is the detuning of the microwave drive with respect to the Larmor frequency. With a measured Rabi frequency of $\Omega_R = 2{,}465(16)$ Hz, we can irradiate the mean Larmor frequency of the two ions, with the difference being about 758 Hz. Thereby, we are able to use a single pulse simultaneously for both ions while accounting for the detuning to achieve a π/2 pulse of 101.1 µs for both ions simultaneously. The corresponding data are shown in Extended Data Fig. 1. The fit includes a shot-to-shot jitter of the microwave offset $\delta\Omega_L$ to account for the uncertainty of the magnetic field. The measurement is performed on a magnetron radius of $r_- = 200(30)$ µm to achieve similar conditions to those in the coupled state.

### Determination of charge-radii differences

We would in principle be able to improve on any charge-radii differences, where this is the limiting factor for the theoretical calculation of $g$. This holds true for most differences between nuclear spin-free isotopes, as well as differences between different atoms, provided they are either light enough for theory to be sufficiently precise or close

enough in nuclear charge $Z$ such that the corresponding uncertainties are still strongly suppressed.

### Fitting function for the Larmor frequency difference

To derive the fitting function of the correlated spin behaviour of the two ions, we first assume that both ions have been prepared initially in the spin-down state, indicated as $\downarrow$. The probability to find each ion individually in the spin-up state ($\uparrow$) then follows the probability of a Rabi oscillation with the frequency of the difference between the ion's Larmor frequency $\omega_{L1}$ or $\omega_{L2}$, respectively, and the common microwave drive frequency $\omega_D$. The probability of finding both ions after the measurement sequence in the spin-up state follows as

$$
\begin{aligned}
P(\uparrow, \uparrow) &= \cos\left(\frac{1}{2}(\omega_{L1} - \omega_D)\tau_{evol}\right)^2 \times \cos\left(\frac{1}{2}(\omega_{L2} - \omega_D)\tau_{evol}\right)^2 \\
&= \left[\frac{1}{2}\left(\cos\left(\frac{1}{2}(\omega_{L1} - \omega_{L2})\tau_{evol}\right) + \cos\left(\frac{1}{2}(\omega_{L1} + \omega_{L2} - 2\omega_D)\tau_{evol}\right)\right)\right]^2.
\end{aligned}
\tag{6}
$$

Similarly, the probability of finding both ions in the spin-down state can be written as

$$
\begin{aligned}
P(\downarrow, \downarrow) &= \sin\left(\frac{1}{2}(\omega_{L1} - \omega_D)\tau_{evol}\right)^2 \times \sin\left(\frac{1}{2}(\omega_{L2} - \omega_D)\tau_{evol}\right)^2 \\
&= \left[\frac{1}{2}\left(\cos\left(\frac{1}{2}(\omega_{L1} - \omega_{L2})\tau_{evol}\right) - \cos\left(\frac{1}{2}(\omega_{L1} + \omega_{L2} - 2\omega_D)\tau_{evol}\right)\right)\right]^2.
\end{aligned}
\tag{7}
$$

Both cases, where either both ions are in the spin-down or the spin-up state have to be considered, as we cannot perform a coherent measurement of the individual Larmor frequencies with respect to the microwave drive. As a result, the information about the Larmor frequency difference is encoded only in the common behaviour of the spins. Therefore, we have to look at the combined probability of both ions either ending up both in the spin-up or spin-down state (case 1; Extended Data Fig. 2), or the complimentary case, where the two spins behave differently, with one ion in the spin-up state and the other ending in the spin-down state. The joint probability is given by

$$
\begin{aligned}
P(t) &= P(\downarrow, \downarrow) + P(\uparrow, \uparrow) = \frac{1}{2}\cos\left(\frac{1}{2}(\omega_{L1} - \omega_{L2})\,t\right)^2 \\
&\quad + \frac{1}{2}\underbrace{\cos\left(\frac{1}{2}(\omega_{L1} + \omega_{L2} - 2\omega_D)\,t\right)^2}_{1/2} \\
&= \frac{1}{4}\cos((\omega_{L1} - \omega_{L2})\,t) + \frac{1}{2},
\end{aligned}
\tag{8}
$$

where, owing to the loss of coherence with respect to the drive frequency, the term in the middle line of equation (8) averages to 1/2. The same formula can be derived for any known initial spin configuration.

### Calculation of the $g$-factor difference

The $g$-factor difference can be directly calculated from the determined relation given by

$$
\Delta g = \frac{2}{\omega_c}\frac{m_e}{m_{ion}}\frac{q_{ion}}{e}\Delta\omega_L.
\tag{9}
$$

Although the input parameters of mass and cyclotron frequency of one of the ions are still required, the precision relative to $\Delta g$ is only of about $7 \times 10^{-5}$, strongly relaxing the need for ultraprecise masses and a cyclotron frequency determination. In contrast, when measuring absolute $g$ factors, the precision of the mass and cyclotron frequency typically limit the achievable precision to the low $10^{-11}$ level.

### Setting constraints on new physics

Measuring the $g$ factor allows for high-precision access to the properties of very tightly bound electrons, and hence to short-range physics, including potential new physics (NP). Bounds on NP can be set with isotopic-shift data on the $g$ factor of hydrogen-like neon. The Higgs-relaxion mixing mechanism, in particular, involves the mixing of a potential new (massive) scalar boson, the relaxion, with the Higgs boson. It has been proposed as a solution to the long-standing electroweak hierarchy problem[13] with the relaxion as a dark-matter candidate[40]. Constraints on this proposed extension of the standard model can be set with cosmological data, as well as with particle colliders, beam dumps and smaller, high-precision experiments (see, for example, ref. [41] and references therein).

The most common approach in atomic physics is to search for deviations from linearity on experimental isotopic-shift data in a so-called King plot analysis[29,35,41–44], which can be a sign of NP, although nonlinearities can also happen within the standard model[12,24,37,44], which limits the bounds that can be set on NP parameters. The King plot approaches also suffer from strong sensitivity on nuclear-radii uncertainties[25]. Here we present constraints on NP from data on a single isotope pair. The influence of relaxions (scalar bosons) on atoms can be expressed[29,43,44] by a Yukawa-type potential (often called the 'fifth force') exerted by the nucleus on the atomic electrons:

$$
V_{HR}(\mathbf{r}) = -\hbar c\,\alpha_{HR}\,A\,\frac{e^{-\frac{m_\phi c}{\hbar}|\mathbf{r}|}}{|\mathbf{r}|},
\tag{10}
$$

where $m_\phi$ is the mass of the scalar boson, $\alpha_{HR} = y_e y_n/4\pi$ is the NP coupling constant, with $y_e$, $y_n$ the coupling of the boson to the electrons and the nucleons, respectively, $A$ is the nuclear mass number and $\hbar$ is the reduced Planck's constant. Yukawa potentials naturally arise when considering hypothetical new forces mediated by massive particles. The corresponding correction to the hydrogen-like $g$ factor is given by[12]

$$
g_{HR} = -\frac{4}{3}\alpha_{HR}\,A\,\frac{(Z\alpha)}{\gamma}\left(1 + \frac{m_\phi}{2Z\alpha m_e}\right)^{-2\gamma} \times \left[1 + 2\gamma - \frac{2\gamma}{1 + \frac{m_\phi}{2Z\alpha m_e}}\right],
\tag{11}
$$

where $\gamma = \sqrt{1 - (Z\alpha)^2}$. The mass scale of the hypothetical new boson is not known[41], apart from the upper bound $m_\phi < 60$ GeV. In the small-boson-mass regime $m_\phi \ll Z\alpha m_e$, the contribution to the $g$ factor simplifies to

$$
g_{HR} = -\frac{4}{3}\alpha_{HR}A\,\frac{(Z\alpha)}{\gamma}, \quad \text{for } m_\phi \ll Z\alpha m_e.
\tag{12}
$$

In the large-boson-mass regime $m_\phi \gg Z\alpha m_e$, we obtain

$$
g_{HR} = -\frac{4}{3}\alpha_{HR}A\,\frac{(Z\alpha)(1 + 2\gamma)}{\gamma}\left(\frac{m_\phi}{2Z\alpha m_e}\right)^{-2\gamma}, \quad \text{for } m_\phi \gg Z\alpha m_e.
\tag{13}
$$

We can set bounds on the NP coupling constant by comparing the measured and calculated values of the $g$-factor isotopic shift (see refs. [12,45] for an implementation of the same idea with transition frequencies in atomic systems). Uncertainties from theory are a source of limitation in this approach. The standard-model contributions to the isotopic shift of the $g$ factor of hydrogen-like neon are given in Extended Data Table 1, as calculated in this work based on the approaches developed in the indicated references. As can be seen, the largest theoretical uncertainty comes from the leading finite nuclear-size correction, and is due to the limited knowledge of nuclear radii (the uncertainty on the finite nuclear-size correction owing to the choice of the nuclear model is negligible at this level of precision). We note that the standard source for these nuclear radii is data on X-ray transitions in muonic atoms[23].

In the NP relaxion scenario, the energy levels of these muonic atoms are also corrected by the relaxion exchange. Another source of r.m.s. charge radii and their differences is optical spectroscopy. The electronic transitions involved are far less sensitive to hypothetical NP than muonic X-ray transitions. The radius difference between $^{20}$Ne and $^{22}$Ne extracted from optical spectroscopy[46] agrees with the one determined from muonic atom data within the respective uncertainties, which shows that NP need not be taken into account to extract nuclear radii from these experiments at their level of precision. To conclude, for our purposes, hypothetical contributions from NP do not interfere with the interpretation of muonic atom data for the extraction of nuclear radii.

Taking $\Delta g_{\text{theo}}^{AA'} = 1.1 \times 10^{-11}$ as the theoretical error on the isotopic shift, it can be seen from equation (12) that this corresponds to an uncertainty of $\Delta y_e y_n \approx 7.1 \times 10^{-10}$ (and a 95% bound on $y_e y_n$ twice as large as this) in the small-boson-mass regime $m_\phi \ll Z\alpha m_e$, which is weaker than the current most stringent bounds coming from atomic physics (H–D 1S–2S, ref. [36]). In the large-boson-mass regime $m_\phi \gg Z\alpha m_e$, our bound remains weaker, but becomes more competitive and is more stringent than those of ref. [35], owing to two favourable factors. First, the nuclear charge $Z$ in equation (13) is larger than the screened effective charge perceived by the Ca$^+$ valence electron, and larger than the charge of the hydrogen nuclei, which also enter the scaling of the bound obtained with these respective ions[29]. Second, when carrying out a King analysis as done in ref. [35], one works with two different transition frequencies, and the leading term in the hypothetical NP contribution in the large-boson-mass regime, which is the equivalent of the right-hand side of equation (13), is cancelled out in the nonlinearity search, owing to its proportionality to the leading finite nuclear-size correction[29], leaving the next term, which scales as $(m_\phi/(2Z\alpha m_e))^{-1-2\gamma}$, as the first non-vanishing contribution.

In the present case, the $g$ factor of a single electronic state is considered (for a single isotope pair), and this cancellation does not occur. This leads to competitive bounds in the large-boson-mass regime with the simple $g$ factor isotopic shift of hydrogen-like ions, as shown in Fig. 3 (where we used the exact result, equation (11)). We compare our bounds on the coupling constant $y_e y_n = 4\pi\alpha_{\text{HR}}$, to the bounds obtained in refs. [35,36], through isotopic-shift measurements in Ca$^+$ (see the curve Ca$^+$ IS-NL in Fig. 3) and H (with nuclear radii extracted from muonic atom spectroscopy), as well as to the bounds obtained through Casimir force measurements[33], globular cluster data[34] and a combination[35]) of neutron scattering[47–50] data and free-electron $g$ factor[1] $(g-2)_e n$.

We also reproduce the preferred range for the coupling constant obtained in ref. [37], through isotopic-shift measurements in Yb$^+$ (Yb$^+$ IS-NL). This range was obtained by assuming that the observed King nonlinearity in the experimental isotopic-shift data is caused by NP. By contrast, all nuclear corrections to the $g$ factor that are relevant at the achieved experimental precision were taken into account in our approach, allowing for an unambiguous interpretation of the experimental data.

In Fig. 3, we also indicate projected bounds that could be obtained from isotopic-shift measurements of the $g$ factors of both hydrogen-like and lithium-like argon. Combining both measurements allows the approximate cancellation of the leading finite nuclear-size corrections through considering a weighted difference[11,27] of hydrogen-like and lithium-like $g$ factors. On the basis of our earlier discussion of the domination of theoretical uncertainties by the uncertainty on the leading finite nuclear-size correction (Table 1), the interest of this approach is readily understood. Our calculations indicate that argon is in the optimal range for setting bounds on $\alpha_{\text{HR}}$ with this approach.

The weighted difference approach is not preferred in the large-boson-mass regime, however, because of strong cancellations of the NP contribution. A similar approach based on a weighted difference of the $g$ factor and ground-state energies of hydrogen-like ions should yield even more stringent bounds[28]. Both these weighted-difference-based approaches are insensitive to uncertainties on the nuclear radii, as such, the bounds that they can generate are fully independent of any assumptions on NP coupling to muons.

## Calculation of the initial phase difference

As our method relies on a single external drive for this specific measurement, used to drive both spins simultaneously, the drive has to be applied at the median Larmor frequency. This results in an additional phase difference that is acquired during the $\pi/2$ pulses. We have determined this phase to be $\Phi_{\text{init}} = 35.8(50)°$ using a numerical simulation. Here we use the knowledge of the Rabi frequency as well as the uncertainty of the magnetic-field determination, which leads to an effective jitter of the microwave drive from cycle to cycle. The simulation is performed for different evolution times, extrapolating to the phase that would be measured for zero evolution time. Although the phase that we can extract from the measured data as a cross-check is consistent with this prediction, we still assign an uncertainty of $\pm 5°$ to the simulation.

## Analysis of systematic shifts of $\Delta g$ of coupled ions

Here we evaluate the total systematic shift and its uncertainty for this method, specifically for the measurement case of $^{20}$Ne$^{9+}$ and $^{22}$Ne$^{9+}$. For this approach, we consider only a separation distance and no common mode. For small common-mode radii $r_{\text{com}} \leq 100$ μm, which we give as an upper limit, the systematic effects discussed here are actually further reduced[20]. We have to consider multiple individual measurements performed with single ions to characterize these frequency shifts and experimental parameters. More explanation on the methods used can be found in ref. [6], and the individual frequency shifts are derived in ref. [51]. We define our electric potential, and specifically the coefficients $C_n$ as

$$\Phi(r, \theta) = \frac{V_r}{2} \sum_{n=0}^{\infty} \frac{C_n r^n}{d_{\text{char}}^n} P_n(\cos(\theta)), \tag{14}$$

with applied ring voltage $V_r$, the characteristic trap size $d_{\text{char}}$ and the Legendre polynomials $P_n$. The magnetic-field inhomogeneities $B_1$ and $B_2$ are defined as

$$\mathbf{B}_1 = B_1\left(z\mathbf{e}_z - \frac{r}{2}\mathbf{e}_r\right), \tag{15}$$

$$\mathbf{B}_2 = B_2\left[\left(z^2 - \frac{r^2}{2}\right)\mathbf{e}_z - zr\mathbf{e}_r\right], \tag{16}$$

where $z$ is the axial position with respect to the electrostatic minimum of the trap. First, we consider the two main axial frequency shifts that depend on the magnetron radius of an ion:

$$\left.\frac{\Delta v_z}{v_z}\right|_{C_4} = -\frac{3}{2}\frac{C_4}{C_2 d_{\text{char}}^2}r_-^2, \tag{17}$$

$$\left.\frac{\Delta v_z}{v_z}\right|_{C_3} = \frac{9}{8}\frac{C_3^2}{C_2^2 d_{\text{char}}^2}r_-^2. \tag{18}$$

If the shift of $v_z$ is measured to be zero for any radius $r_-$, these two shifts cancel and we can conclude that $C_4 = \frac{3}{4}\frac{C_3^2}{C_2}$. As it is typically not feasible to tune this for arbitrary radii, especially as higher orders will have to be considered as well for larger radii, we allow a residual $\eta_{\text{el},r_-}$, which includes both the residual observed shift and all neglected smaller contributions. This is a relative uncertainty, scaling with $r^2$:

$$\left.\frac{\Delta v_z}{v_z}\right|_{\text{el}} = \frac{9}{8}\frac{C_3^2}{C_2^2 d_{\text{char}}^2}r_-^2 - \frac{3}{2}\frac{C_4}{C_2 d_{\text{char}}^2}r_-^2 = \eta_{\text{el},r_-} \tag{19}$$

Similarly, we consider all frequency shifts that depend on the cyclotron radius $r_+$ of an ion:

$$\left.\frac{\Delta v_z}{v_z}\right|_{C_4} = -\frac{3}{2}\frac{C_4}{C_2 d_{\mathrm{char}}^2}r_+^2, \tag{20}$$

$$\left.\frac{\Delta v_z}{v_z}\right|_{C_3} = \frac{9}{8}\frac{C_3^2}{C_2^2 d_{\mathrm{char}}^2}r_+^2. \tag{21}$$

The electrostatic contributions are identical to those for the magnetron mode, and per the assumption above will also combine to the same $\eta_{\mathrm{el},r_+}$, scaling with the cyclotron radius. However, we have to consider the additional terms that stem from the magnetic-field inhomogeneities, which are sizeable in this mode owing to the significantly higher frequency:

$$\left.\frac{\Delta v_z}{v_z}\right|_{B_2} = \frac{B_2}{4B_0}\frac{v_+ + v_-}{v_+ v_-}v_+ r_+^2$$
$$\approx \frac{B_2}{B_0}\frac{v_+^2}{2v_z^2}r_+^2, \tag{22}$$

$$\left.\frac{\Delta v_z}{v_z}\right|_{B_1} = -\frac{3B_1 C_3 v_c v_+}{4B_0 C_2 d_{\mathrm{char}} v_z^2}r_+^2$$
$$\approx -\frac{3B_1 C_3 v_+^2}{4B_0 C_2 d_{\mathrm{char}} v_z^2}r_+^2. \tag{23}$$

In addition, for large cyclotron excitations, we have to consider the relativistic effect of the mass increase, which also slightly shifts the axial frequency:

$$\left.\frac{\Delta v_z}{v_z}\right|_{\mathrm{rel.}} = -\frac{3B_1 C_3 v_c v_+}{4B_0 C_2 d_{\mathrm{char}} v_z^2}r_+^2 \tag{24}$$

The combined shift depending on magnetic inhomogeneities can be expressed as

$$\left.\frac{\Delta v_z}{v_z}\right|_{\mathrm{mag}} = \left(\frac{B_2}{B_0}\frac{v_+^2}{2v_z^2} - \frac{3B_1 C_3 v_+^2}{4B_0 C_2 d_{\mathrm{char}} v_z^2}\right)r_+^2 = \eta_{\mathrm{mag}}. \tag{25}$$

Although we cannot currently tune these contributions actively (which could be implemented by using active compensation coils[22]), we can slightly shift the ion from its equilibrium position to a more preferable position along the $z$ axis to minimize the $B_2$ coefficient. Doing so, we have achieved frequency shifts of $v_z$ close to zero for any cyclotron excitations as well, which means these terms have to cancel as well. We will still allow for another residual error from higher orders, as well as a small residual shift, defined as $\eta_{\mathrm{mag}}$. The observed difference in the frequency shift between cyclotron and magnetron excitations $\eta_{\mathrm{mag}} + \eta_{\mathrm{el},r_+} - \eta_{\mathrm{el},r_-}$ can be used to cancel the identical electric contributions $\eta_{\mathrm{el},r_+}$ and $\eta_{\mathrm{el},r_-}$ when measuring at the same radius. If we solve this combined equation for $C_3$, we are left with only the magnetic-field-dependent terms $B_1$ and $B_2$, which is what the Larmor frequency difference is sensitive to

$$C_3 = \frac{2}{3}\frac{B_2 C_2 d_{\mathrm{char}}}{B_1} - \underbrace{\frac{4}{3}\frac{B_0 C_2 d_{\mathrm{char}} v_z^2}{B_1 v_+^2 r_+^2}\eta_{\mathrm{mag}}}_{\xi}$$
$$= \frac{2}{3}\frac{B_2 C_2 d_{\mathrm{char}}}{B_1} - \xi, \tag{26}$$

where $\xi$ summarizes the shifts depending on the radial modes of the ion. Now, instead of looking at frequency shifts of individual ions, we consider the effects on coupled ions. Owing to their mass difference, the coupled state is not perfectly symmetrical but slightly distorted owing to the centrifugal force difference. In the case of the neon

isotopes, this leads to a deviation of $\delta_{\mathrm{mag}} = 0.87\%$, with the definition of $r_1 = d_{\mathrm{sep}}\frac{(1-\delta_{\mathrm{mag}})}{2}$ and $r_2 = d_{\mathrm{sep}}\frac{(1+\delta_{\mathrm{mag}})}{2}$, when choosing ion 1 to be ${}^{20}\mathrm{Ne}^{9+}$ and ion 2 as ${}^{22}\mathrm{Ne}^{9+}$. Consequently, the frequency difference $v_{L_1} - v_{L_2}$ will be positive, as the $g$ factor (and therefore the Larmor frequency) for ${}^{20}\mathrm{Ne}^{9+}$ is larger than for ${}^{22}\mathrm{Ne}^{9+}$. We now consider the axial position shift as a function of the slightly different $r_-^2$. This is given by

$$\Delta z = \frac{3}{4}\frac{C_3}{d_{\mathrm{char}}C_2}r_-^2. \tag{27}$$

Now we want to express all frequency shifts in terms of $v_L$, which is to a very good approximation dependent on only the absolute magnetic field, first considering only the effect of $B_1$ and all shifts along the $z$ axis:

$$\left.\frac{\Delta v_L}{v_L}\right|_{B_1} = \Delta z\frac{B_1}{B_0}. \tag{28}$$

The difference in the shift for the individual ions can then be written as

$$\left.\frac{\Delta(\Delta v_L)}{v_L}\right|_{B_1} = \frac{\Delta v_{L_1} - \Delta v_{L_2}}{v_L}$$
$$= (\Delta z_1 - \Delta z_2)\frac{B_1}{B_0}$$
$$= \frac{3}{4}\frac{C_3}{C_2}\frac{B_1}{B_0 d_{\mathrm{char}}}(r_1^2 - r_2^2)$$
$$= \left(\frac{1}{2}\frac{B_2}{B_0} - \frac{3}{4}\frac{B_1\xi}{B_0 C_2 d_{\mathrm{char}}}\right)(r_1^2 - r_2^2)$$
$$=: v_{L,B_1}^{\mathrm{rel}}. \tag{29}$$

We have now the additional uncertainties all summarized in the term scaling with the above-defined factor $\xi$. The final shift to consider is the same radial difference as mentioned before in the presence of $B_2$. This leads to additional individual shifts in the $v_L$ of the ions as

$$\left.\frac{\Delta v_L}{v_L}\right|_{B_2} = \frac{-B_2}{2B_0}r^2. \tag{30}$$

As a relative shift with respect to the measured Larmor frequency difference, this can be written as

$$\left.\frac{\Delta(v_L)}{\Delta v_L}\right|_{B_2} = \frac{\Delta v_{L_1} - \Delta v_{L_2}}{v_L}$$
$$= -\frac{1}{2}\frac{B_2}{B_0}(r_1^2 - r_2^2)$$
$$=: v_{L,B_2}^{\mathrm{rel}}. \tag{31}$$

Combining these shifts, $v_{L,B_2}^{\mathrm{rel}}$ and $v_{L,B_1}^{\mathrm{rel}}$, results in

$$\frac{\Delta(\Delta v_{L,\mathrm{tot}})}{v_L} = v_{L,B_1}^{\mathrm{rel}} + v_{L,B_2}^{\mathrm{rel}}$$
$$= \left[\frac{1}{2}\frac{B_2}{B_0} - \frac{3}{4}\frac{B_1\xi}{B_0 C_2 d_{\mathrm{char}}} - \frac{1}{2}\frac{B_2}{B_0}\right](r_1^2 - r_2^2)$$
$$= -\frac{3}{4}\frac{B_1}{B_0 C_2 d_{\mathrm{char}}}\xi(r_1^2 - r_2^2)$$
$$= -\frac{3}{4}\frac{B_1}{B_0 C_2 d_{\mathrm{char}}}\frac{4}{3}\frac{B_0 C_2 d_{\mathrm{char}} v_z^2}{B_1 v_+^2 r_+^2}\eta_{\mathrm{mag}}(r_1^2 - r_2^2)$$
$$= -\frac{v_z^2}{v_+^2}\frac{\eta_{\mathrm{mag}}}{r_+^2}(r_1^2 - r_2^2)$$
$$= 6 \times 10^{-13}. \tag{32}$$

We find that, in the ideal case where neither magnetron nor cyclotron excitations produce shifts of the measured axial frequency $\nu_z$, the final difference of the Larmor frequency is also not shifted at all. Here we use the worst case, with a measured combined relative shift for $\frac{\eta_{mag}}{r_+^2} \approx \frac{125\,\text{mHz}}{560}$. This corresponds to a systematic shift of $\frac{\Delta(\Delta\nu_{L,tot})}{\Delta\nu_{L,tot}} = 6 \times 10^{-13}$, which we did correct for in the final result. This has been confirmed by performing two measurements on different separation distances, of $d_{sep} = 340\,\mu\text{m}$ and $d_{sep} = 470\,\mu\text{m}$. Both measurements have been in agreement after correcting for their respective expected systematic shift. The uncertainty of this correction of $5 \times 10^{-13}$ has been evaluated numerically by combining the uncertainties of $\eta_{mag}$ and the radii intrinsic to its determination, an uncertainty of $\delta_{mag}$ and the potential of a systematic suppression of the systematic shift by a residual common-mode radius.

### Different axial amplitudes

The measurement is performed by first thermalizing the $^{20}\text{Ne}^{9+}$, then increasing the voltage to bring the $^{22}\text{Ne}^{9+}$ into resonance with the tank circuit. This will slightly decrease the axial amplitude of the $^{20}\text{Ne}^{9+}$, which nominally has the larger amplitude when cooled to the identical temperature, compared at the same frequency owing to its lower mass. The residual difference in amplitude will lead to a further systematic shift in the presence of a $B_2$, which has been evaluated to about $3 \times 10^{-14}$ and can therefore safely be neglected at the current precision.

### g-factor calculation

In Extended Data Table 1, the individual contributions to the $g$ factors of both ions are shown. The main uncertainty, the higher-order two-loop QED contribution, is identical for both ions and does cancel in their difference and can be neglected for the uncertainty of $\Delta g$. The finite nuclear size (FNS) correction gives the dominant uncertainty in $\Delta g$, which in turn is determined by the uncertainty of the r.m.s. radius[23]. The next error comes from the nuclear polarization correction, which sets a hard limit for a further improvement in the determination of the r.m.s. radius. The difference in the spectra of photonuclear excitations of $^{20}\text{Ne}$ and $^{22}\text{Ne}$ defines the contribution of the nuclear polarization to $\Delta g$. As the dominant contribution to the nuclear polarization of $^{20,22}\text{Ne}$ comes from the giant resonances, one has to estimate the isotope difference of this part of the spectrum. The measurements of the absolute yields of the various photonuclear reactions are reported in refs. [52,53] for $^{20}\text{Ne}$ and in refs. [54,55] for $^{22}\text{Ne}$. On the basis of these data, we conclude that the integrated cross-section for the total photoabsorption between $^{20}\text{Ne}$ and $^{22}\text{Ne}$ differs by less than 20%, which we take as the relative uncertainty of the nuclear polarization contribution to $\Delta g$. The hadronic vacuum polarization (see, for example, ref. [56]) corresponds to the small shift of the $g$ factor by the virtual creation and annihilation of hadrons and is largely independent of the nuclear structure. In the $g$-factor difference of $^{20}\text{Ne}^{9+}$ and $^{22}\text{Ne}^{9+}$, the QED contribution to the nuclear recoil can be resolved independently from all common contributions. A test of this contribution by means of an absolute $g$-factor measurement is possible for only the small regime from carbon to silicon and for only stable isotopes without nuclear spin. For smaller $Z \leq 6$, the QED contribution is too small to be resolved experimentally, and for $Z > 14$, the uncertainty of the two-loop QED contribution is too large to test the QED recoil. In addition, such an absolute $g$-factor measurement would also require the ion mass to similar precision, which is not the case for the approach via the direct difference measurement performed here.

### Data availability

The datasets generated during and/or analysed during the current study are available from the corresponding author on reasonable request.

### Code availability

All code used for the analysis and production of results of the current study are available from the corresponding author on reasonable request.

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

**Acknowledgements** This work was supported by the Max Planck Society (MPG), the International Max Planck Research School for Quantum Dynamics in Physics, Chemistry and Biology (IMPRS-QD), the German Research Foundation (DFG) Collaborative Research Centre SFB 1225 (QUANT) and the Max Planck PTB RIKEN Center for Time, Constants, and Fundamental Symmetries. This project has received funding from the European Research Council (ERC) under the European Union's Horizon 2020 research and innovation programme under grant agreement number 832848 FunI. A.V.V. acknowledges financial support by the Priority 2030 Federal Academic Leadership Program and by the Government of the Russian Federation through the ITMO Fellowship and Professorship Program. This work comprises parts of the PhD thesis work of C.K., T.S. and J.M. to be submitted to Heidelberg University, Germany.

**Author contributions** The experiment was maintained and performed by T.S., B.T., C.K., F.H., J.M. and S.S. The data were analysed by T.S., F.H. and S.S. The manuscript was written by T.S. Theoretical calculations were performed by A.V.V., V.D. and Z.H. All authors discussed and approved the data as well as the manuscript.

**Funding** Open access funding provided by Max Planck Society.

**Competing interests** The authors declare no competing interests.

**Additional information**
**Correspondence and requests for materials** should be addressed to Tim Sailer.

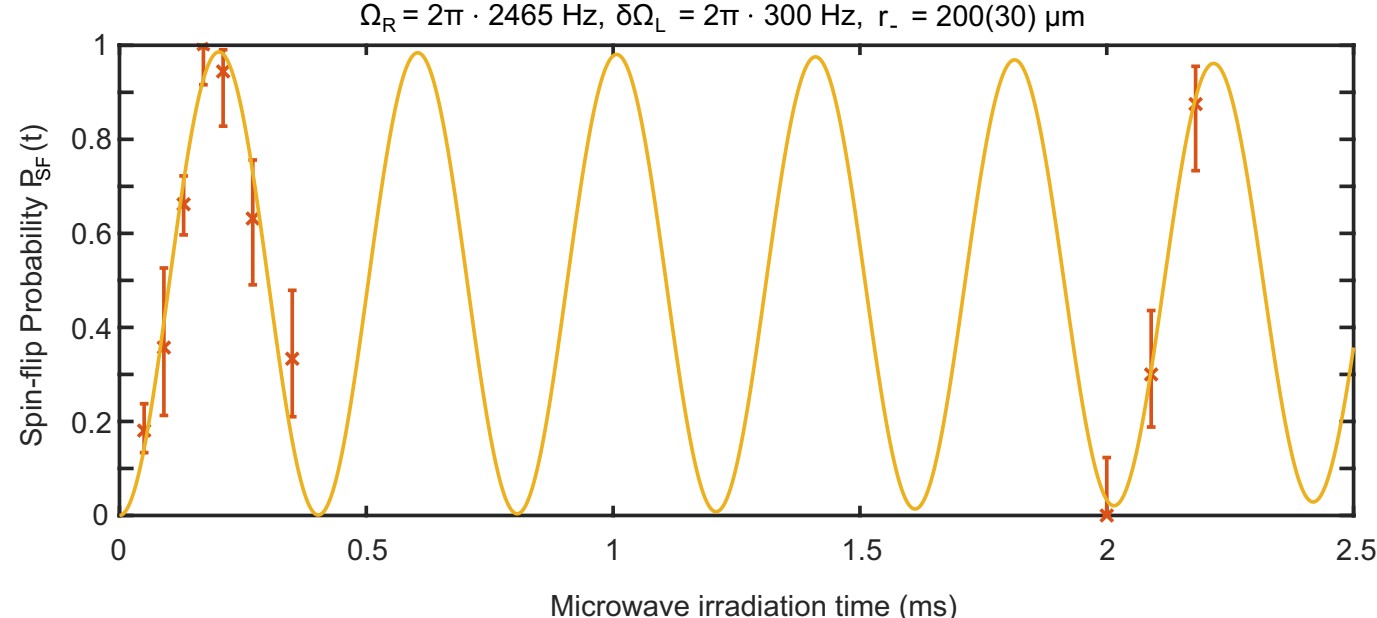

**Extended Data Fig. 1 | The measured Rabi frequency $\Omega_R$ on a single ion.** The ion is excited to a magnetron radius of $r_- = 200(30)$ µm, to ensure similar conditions as in the coupled state. The probability of inducing a change of spin orientation $P_{SF}(t)$ is modulated by the pulse length of the microwave irradiation time. The fit includes a varying microwave offset $\delta\Omega_L$ to account for the uncertainty of the magnetic field determination. Error bars represent the 68% CI.

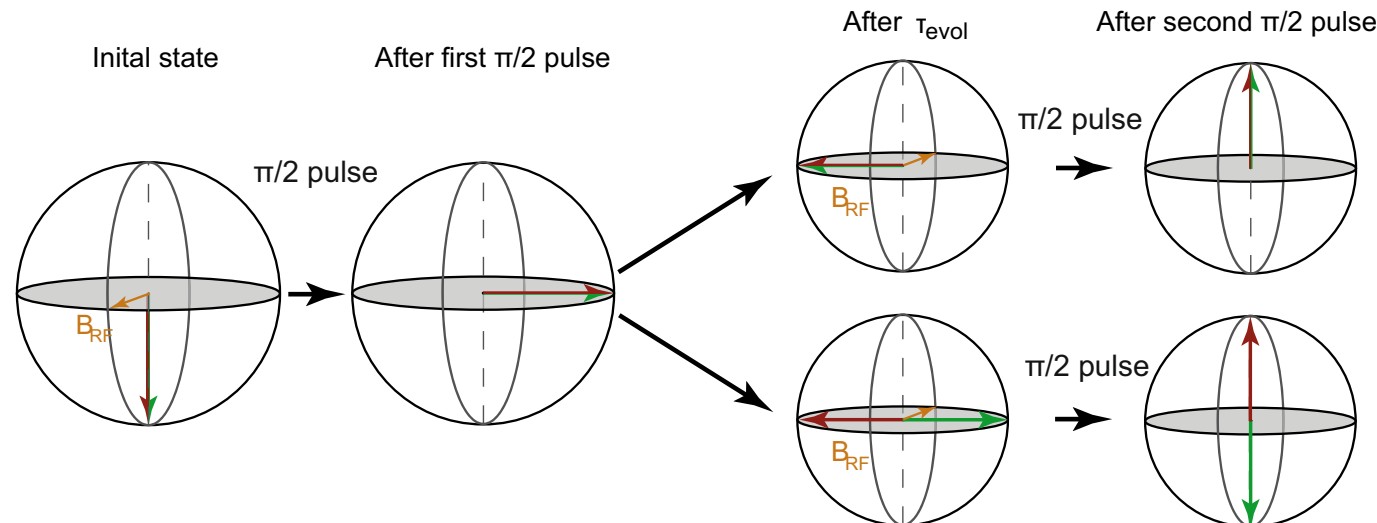

**Extended Data Fig. 2 | Bloch sphere representation of two of the possible outcomes of the measurement for an initial configuration with both ions in spin-down state.** After both spins are rotated around the applied drive vector (orange) to the equatorial plane ($\pi/2$ pulse), they precess freely for the evolution time $\tau_{evol}$. As the coherence with the applied drive is lost, the phase of the second time the drive is applied is completely random, leading to a reduction in visibility. The relative phase of the spins with respect to each other is encoded in their probability to behave identically, maximised when they are in phase (upper scenario) and minimal when their phase difference is 180°.

**Extended Data Table 1 | Contributions to the calculation of the *g*-factors of $^{20}$Ne$^{9+}$ and $^{22}$Ne$^{9+}$ and their difference and the final experimental result**

| | $^{20}$Ne$^{9+}$ | $^{22}$Ne$^{9+}$ | |
|---|---|---|---|
| Dirac value (point nucleus) | 1.996 445 170 898(2) | 1.996 445 170 898(2) | |
| Finite nuclear size, FNS | 0.000 000 004 762(7) | 0.000 000 004 596(12) | |
| QED, one loop ($\alpha$) | 0.002 325 473 294(1) | 0.002 325 473 294(1) | |
| QED, two loop ($\alpha$)$^2$ | $-$0.000 003 547 780(117) | $-$0.000 003 547 780(117) | |
| QED, $\geq$ three loop ($\alpha$)$^{3+}$ | 0.000 000 029 524(1) | 0.000 000 029 524(1) | |
| Recoil | | | |
| Non-QED | 0.000 000 146 093 420 | 0.000 000 132 810 693 | |
| QED | 0.000 000 000 477 954(1) | 0.000 000 000 434 499(1) | |
| ($\alpha/\pi$)($m_e/M$) | $-$0.000 000 000 113 2(6) | $-$0.000 000 000 102 9(5) | |
| ($m_e/M$)$^2$ | $-$0.000 000 000 044 1(2) | $-$0.000 000 000 036 5(2) | |
| Hadronic vacuum pol. | 0.000 000 000 003 36(3) | 0.000 000 000 003 36(3) | |
| Nuclear polarization | $-$0.000 000 000 001 9(9) | $-$0.000 000 000 002 0(10) | |
| g factor total theory | 1.998 767 277 112(117) | 1.998 767 263 638(117) | |
| Difference (in $10^{-9}$) | | | Ref. |
| FNS | | 0.166(11) | TW, [57] |
| Recoil, non-QED | | 13.2827 | [58] |
| Recoil, QED | | 0.0435 | [10] |
| Recoil, ($\alpha/\pi$)($m_e/M$) | | $-$0.0103 | [59] |
| Recoil, ($m_e/M$)$^2$ | | $-$0.0077 | [59] |
| Deformation | | < 0.0001 | [60] |
| Polarization | | 0.0001(3) | TW |
| $\Delta$g Total theory | | 13.474(11)$_{FNS}$ | |
| $\Delta$g Experiment | | 13.47524(53)$_{stat}$(99)$_{sys}$ | |

Each QED contribution in $\alpha^n$ is calculated in orders of $(Z\alpha)^n$, scaling identically for both isotopes. The value of the fine-structure constant used in the calculation is $\alpha^{-1}$=137.03599911. $m_e$ and $M$ are the mass of the electron and the nucleus, respectively. Values presented without uncertainties are exact to all given digits[10, 57–60]. TW, this work.