## [Peer Review File · Nature]

Manuscript Title: Measurement of the bound-electron g -factor difference in coupled ions

Reviewer Comments & Author Rebuttals

Reviewer Reports on the Initial Version:

Referee #1 (Remarks to the Author):

REVIEW OF 'DIRECT BOUND-ELECTRON g FACTOR DIFFERENCE MEASUREMENT WITH COUPLED IONS

This paper reports the results of a very impressive experiment, and should definitely be published. The general field of the experimental study of g factors of single-electron ions with high precision is a very important one, as it has been possible to develop the corresponding theory to a similar level of precision. The most accurate determination of the electron mass, for example, comes from this area. The experiment described in the present paper extends the field significantly. By considering instead of individual g factors their difference in two isotopes of neon, the largest theory contributions cancel, so that new facets of the field can be studied. The experiment is a *tour de force*, and has so many interesting features that just its description would be publishable. But because of the additional theoretical control present, the paper has three additional important impacts on physics. The first is the test of the very non-trivial recoil calculations required to go beyond Furry representation. The second is the indirect determination of the charge radii difference of the two nuclei. While the charge radii can be obtained in other ways, the accuracy is limited, and if one takes the results of the paper assuming the recoil theory is correct, a much more accurate result is found. Finally, as with any precision test, exotic theories that would lead to a discrepancy between the predictions of the standard model (QED in this case) and experiment can be ruled out. The paper is very carefully written, except for line 316, where the word 'to' is left out, I only have some writing suggestions the authors should feel free to disregard.

In the second sentence of the abstract, the use of the word 'specifically' could be taken to mean that the excellent agreement of QED theory with experiments, which applies to a wide range of physical situations, is confined to g -factor measurements: if 'In particular' were used instead of 'Specifically', the meaning would be clearer.

'The difference of isotopes' (line 5) needs what kind of difference specified to be meaningful: writing 'Measuring the g factor in different isotopes' would resolve this.

'Do not have to be considered' (line 7) could be replaced by simply 'cancel'.

The comma in line 28 is not needed.

While 'unpractical' (line 159) is acceptable, 'impractical' or 'impracticable' are closer to describing experimental challenges.

Referee #2 (Remarks to the Author):

A. Summary of the key results

The authors have performed for the first time a differential bound-state electron g factor measurement by storing two neon isotopes simultaneously in a Penning trap system. This method vastly reduces the influence of fluctuations of experimental parameters, such as the magnetic field, compared to subsequent measurements on individual ions. Along with this measurement, the authors provide computations of higher order nuclear recoil effects, and establish agreement between experiment and theory at the 10^{-11} level. The limiting factor at this point is the insufficient knowledge of the difference in nuclear charge radius between the two isotopes. Experiment itself would already be 10 times better. Reversing the argument, assuming correctness of theory, a new, improved nuclear charge radius difference can be established. In addition, additional contributions to the electron-nucleus interaction via 'new physics' such as bosons related to a Higgs portal can be excluded.

B. Originality and significance: if not novel, please include reference

The presented experimental work is astonishing and a delight to read. This is absolutely first-rate work, original, and significant in the sense that a differential measurement can eliminate external disturbances, in particular magnetic field fluctuations. In addition, the technique demonstrates incredible levels of control over individual ions. As discussed in the paper, this should lead to significant, further improvements in the future. This outstanding experimental work is accompanied by solid theoretical support work.

C. Data & methodology: validity of approach, quality of data, quality of presentation

The validity of the approach and the quality of the data are clearly demonstrated and require in my view no further comment. The quality of the presentation is also high. The manuscript is well written and essentially error free, be it typographic or content wise. I do have a list of suggestions for improvements and corrections in terms of presentation. I will list them at the very bottom to keep them out of the scientific discussion.

D. Appropriate use of statistics and treatment of uncertainties

Generally, the analysis looks solid, but I have questions concerning the uncertainties of theory values given in Table 2 of 3.4, and also the presentation of those numbers in the main paper.

Table 2: some of the values, such as recoil, Non-QED, QED, and hadronic effects are given without uncertainties. If we can trust that they are good to the last digit, this should not be a problem for the current purpose, but this should be explicitly stated. Also, why are the hadronic effects identical, if the nucleon composition of these two isotopes differs not insignificantly? For the current purpose, they only need to cancel to 30% to leave the results unaffected. However, such effects are notorious (similar to nuclear polarization) as they tend to present a "hard wall". It would be good to know what kind of limit they pose. Because that could severely hamstring future improvements with this

method.

Table 1: The theory values for the recoils are cut off at 3 digits behind the decimal point, with no uncertainties given. For the comparison between theory and experiment, this is of course sufficient because of the dominant uncertainty from FNS. However, when the argument is turned around (line 310), this appears insufficient. The authors state that in this case (assuming the theory is sufficiently trusted), the value for the charge radius difference can be improved by approximately 1 order of magnitude. Since the previously measured value leads to an uncertainty of 0.011 for the FNS, a 10x improvement would nail down the FNS to 0.001 (assuming FNS uncertainty goes linear with charge radius uncertainty). At that point, the other theory values in Table 1 would be too inaccurate, as they have to be at least +/- 1 digit. While better numbers are found in the supplementary materials, I believe that the main paper must be able to survive standalone. I suggest to add one more digit in Table 1. In this context, it is also apparent that the cancellation of hadronic effects between the isotopes should be at least 10% or better. That should be motivated.

E. Conclusions: robustness, validity, reliability

The conclusions in terms of the theory/experiment comparison and the ability to improve charge radius differences with this technique are sound.

I have concerns about the 'new' physics section though. Overall, again, the motivation is sound, and I have no issues with the findings in terms of the (TW) curve shown in Figure 3. I also greatly appreciate this approach in the sense that it eliminates this more than awkward reliance on the linearity of the King Plot (in this context, the authors could also cite arXiv:2111.01429v1 (Figueroa et al), which weakens the Yb case from Counts et al.).

However, the section on 'new' physics in the main manuscript is exceedingly short and gives few details. It gives the feeling of a bolt-on section. Important information like what the other curves represent, should in my opinion be addressed in the main body, not just the supplement. If this is an issue of a page limit, I suggest that the description of the experimental method be shortened to create space to work this out in better detail.

Finally: If the reader is assumed to take the H-D 1s-2s and $(g-2)_n$ curves seriously, then the manuscript makes no attempt to justify why the current result would be considered competitive. Everywhere in this graph, TW seems to be at least 1.5 orders of magnitude weaker than the best limit (this was already the problem in Counts' paper). In the supplement the authors comment briefly on this fact, but then quickly deflect to discussing the advantages over the King Plot methods, which also don't seem to be competitive. While the statement in the abstract "allows setting constraints" is technically not wrong, this is normally assumed to mean "allows setting new best constraints" in the Beyond-the-Standard-Model community. At the minimum, the result should lend strong support to an existing limit. To me, the TW curve does not seem to do that. However, if the authors could convincingly argue that there could be issues with H-D 1s-2s and/or $(g-2)_n$, or that these limits have some model-dependencies built in that could potentially end up weakening them (say, reliance on a certain proton radius value?), then possibly a case could be made.

F. Suggested improvements: experiments, data for possible revision

I have no suggested improvement on the experiment itself, nor the data.

G. References: appropriate credit to previous work?

Yes.

H. Clarity and context: lucidity of abstract/summary, appropriateness of abstract, introduction and conclusions

See my comment above about setting constraints. In particular in the abstract, this would be misunderstood by readers as setting improved limits.

Summary: This is spectacular experimental work of the highest calibre. I could support publication in Nature on this aspect alone, once the 'new' physics section is updated. But at this point, the Higgs portal section is not compelling, and by itself does not make a case for Nature.

List of suggestions:

L3: "the electron's magnetic moment"

L6: only relatively benign nuclear properties come into play, so I'm not sure whether "intricate" is the right word here.

L19: see comments above about what the expectation for "setting constraints" is.

Table 1: FNS should be written out or defined in the caption or main text before it is used here.

Table 1: I suggest adding one digit to the recoil numbers.

L121-126: Seems to me that this is not a valid sentence, structure-wise.

L167: This is the first time that timing is mentioned. Best to say "this takes about 10 min", especially since this is later done repeatedly.

Figure 1: "Trap centre" looks not well defined. Maybe it could have a dot assigned to it, just like r_{com} , with another arrow between them.

Figure 2: the grey box could use a somewhat lighter grey

Figure 2: the green data point in the lower plot is very hard to spot, make it larger, maybe a filled circle instead of a cross.

L316: see above about "setting constraints".

Figure 3: the caption or the main text should refer to the other experiments, at least the ones that actually seem to set stronger limits.

References:

L440: this does not look like an arXiv citation.

Supplement:

L765: is the correct citation for the $(g-2)_n$ work really reference 19, Solaro, which is on the Ca+ King Plot work, or is that a mistake?

Referee #3 (Remarks to the Author):

A. Summary of key results:

The key result presented in this paper is a direct measurement of the difference between the g factors of $^{20}\text{Ne}^{9+}$ and $^{22}\text{Ne}^{9+}$ with an uncertainty that is 2 orders of magnitude more precise than any previous such comparison. The uncertainty in the experimental value is a factor of 10 smaller than the uncertainty in the theoretical value. As such, this measurement enables, for the first time, the ability to check the QED part of the nuclear recoil contribution to the g factor. This comparison provides a validation of the theoretically calculated value. Finally, agreement between experiment and theory enables constraints to be placed on searches for physics beyond the standard model.

B. Originality and significance

The paper presents for the first time the description of a new experimental technique for g factor measurements on highly charged ions. The technique is certainly novel and enabled a significant increase in precision in the measurement that was performed compared to previous techniques. This led to the ability to perform a new experimental check of terms in the QED calculations.

The experimental technique described in this paper does combine the previously reported continuous Stern-Gerlach effect measurement technique used at ALPHA TRAP and elsewhere, and the two-ion technique (with ions on a phase locked common magnetron orbit) developed at MIT for precise mass measurements. However, this is much more significant than an incremental development. The two-ion technique was developed over 20 years ago, but was only very recently re-implemented with the MIT trap at Florida State University (PRL 127, 243001 (2021)). This is also the first time this technique has been demonstrated for highly-charged ions, and for ions of different nominal mass number. The spectroscopy technique for inducing spin flips also used a Ramsey type technique, which is an additional and significant development compared to previous g factor measurements and the spin flip measurement technique is used here for the first time on two ions simultaneously. These techniques along with those to manipulate and combine the ions in the two traps in these very precise ways are extremely sophisticated and impressive. In my opinion, the experimental techniques and the results that they have enabled do warrant publication in Nature.

C. Data & methodology

The paper includes graphs in figure 2 showing the data obtained for the combined spin transition probability for the individual measurements described in the paper. The text of the paper provides a

clear description of how these data are obtained and then how they are used to obtain the final Lamor frequency difference.

D. Appropriate use of statistics and treatment of uncertainties

The authors use appropriate statistical methods to determine their final statistical uncertainty. The authors also give a thorough description of the potential systematic shifts and uncertainties in the experiment.

However, I do find a slightly different number for the systematic uncertainty in the final result if I follow the numbers in the paper:

Eqn (29) states a relative uncertainty in the Lamor frequency difference of 6×10^{-13} (incidentally this differs from the definition on line 601). If I multiply 6×10^{-13} by 112 GHz and divide by the Lamor frequency difference of 758.75 Hz, I get 8.8×10^{-5} for the fractional uncertainty in the g factor difference, corresponding to 0.00119×10^{-9} rather than 0.00099 for the systematic uncertainty. I guess this could be a rounding error?

E. Conclusions

The conclusions that the authors draw from their results are valid and justified by the data. The authors provide a promising, but realistic outlook of future measurements that could be performed using this technique and their implications.

F. Suggested improvements

One thing that was not obvious to me at first, and that could be emphasized in the paper is the fact that this technique significantly reduces the demand on the precision of the cyclotron frequency measurement and that only the cyclotron frequency of one ion needs to be measured at all to obtain the difference in g. This is stated somewhat passively in the first sentence of section 3.3.

I say this because on lines 148-155, the authors emphasize how much a direct measurement of g is limited by the cyclotron frequency measurement, and also the masses of the ion and the electron. Perhaps providing the formula for Δg (in the methods section) would help make this clearer. See other specific comments in section H.

G. References

Overall, the references are appropriate.

On line 662, please provide a reference for the “double dip cyclotron frequency measurement technique”.

H. Clarity and context

I list here a few specific comments, questions and suggestions:

- Line 41-45: “However, studying these effects...(see Methods 2)”.

Do the authors mean Methods 3.4, g factor calculation?

- Line 55-58: “For the calculated difference $\Delta g = \dots$ the absolute uncertainty is decreased by two orders of magnitude compared to the absolute values...”

Perhaps I am not interpreting this statement correctly or missing something. The absolute uncertainty in Δg is 0.011×10^{-9} . What do the authors mean by “compared to the absolute values”? I

took this to mean the uncertainty in the absolute g factor values, which is 1.17×10^{-10} . The ratio of these two numbers is one order of magnitude, not two.

- Table 1: The abbreviation FNS is used here. It is defined in the Methods section, but not here or in the main text.

- Line 121 – 126: “Additionally, the presence of ... with the Lamor frequency $\nu_L = \dots$ [that] amounts to about 112 GHz, ...”

- Line 175 – 179: “The combined motion, ... around a common guiding center on a radius r_{com} and a rotation of this guiding center around the trap center.” This sentence as written is not completely clear.

Suggestion: “The combined motion, ... around a common guiding center, that rotates around the trap center on a radius r_{com} .”

- Line 187 – 191: Subsequently we are able to convert common mode into separation mode radius...”

The language used here is a little loose. As I understand it, it is the classical action that is transferred between normal modes of an ion in the Penning trap. Is this also the case for the separation and common mode? Is there a one-to-one correspondence between the radius of common and separation modes after this conversion?

- Line 193 – 196: “We use these tools to prepare the ions with ... $d_{\text{sep}} = 400 \text{ um}$ and a comparably small common mode radius r_{com} .”

Is the common mode radius not made to be approximately equal to 0?

- Line 586: “...larger than for $^{22}\text{Ne}^{9+}$ [.]”

- Line 602: “This [has] been confirmed...”

- Line 628: “inferred, without ... common mode [however].” Remove however.

- Line 670: “The strong magnetic bottle...gives rise to a force [that is] dependent on...”

- Organization of Methods section. The structure of the Methods section is not entirely intuitive, e.g. Section 2 is titled “Calculation of the initial phase”, but should really be the first sub-section of section 2, because subsection 2.1 is “Combined systematic shifts...”, which isn’t a subsection under the main heading of section 2. The same applies to subsection 2.2.

Section 3 “Ion manipulation” is appropriate for subsections 3.1, 3.2 and 3.3, but subsection 3.4 “g factor calculation” doesn’t fit under this section heading. The same is true for 3.5.

Author Rebuttals to Initial Comments:

Concerning the comments of the Referees, please find the answers attached in line.

Referee #1:

In the second sentence of the abstract, the use of the word 'specifically' could be taken to mean that the excellent agreement of QED theory with experiments, which applies to a wide range of physical situations, is confined to g-factor measurements: if 'In particular' were used instead of 'Specifically', the meaning would be clearer.

We agree, this has been changed.

'The difference of isotopes' (line 5) needs what kind of difference specified to be meaningful: writing 'Measuring the g factor in different isotopes' would resolve this.

This was originally meant to cover a broader field than only g factors, the same would be true for many different isotope shift measurements. While we do not insist on the exact formulation, this provides a link to other similar measurements.

'Do not have to be considered' (line 7) could be replaced by simply 'cancel'.

Agreed, simpler and shorter, this has been changed.

The comma in line 28 is not needed.

Deleted.

While 'unpractical' (line 159) is acceptable, 'impractical' or 'impracticable' are closer to describing experimental challenges.

Changed to impractical.

Thank you for the suggestions and positive feedback.

Referee #2:

Comments:

"Generally, the analysis looks solid, but I have questions concerning the uncertainties of theory values given in Table 2 of 3.4, and also the presentation of those numbers in the main paper.

Table 2: some of the values, such as recoil, Non-QED, QED, and hadronic effects are given without uncertainties. If we can trust that they are good to the last digit, this should not be a problem for the current purpose, but this should be explicitly stated. Also, why are the hadronic effects identical, if the nucleon composition of these two isotopes differs not insignificantly? For the current purpose, they only need to cancel to 30% to leave the results unaffected. However, such effects are notorious (similar to nuclear polarization) as they tend to present a "hard wall". It would be good to know what kind of limit they pose. Because that could severely hamstring future improvements with this method.

We agree with the Referee and have added the uncertainties for the terms "QED, \geq three loop", "Nuclear recoil QED", and "Hadronic effects". We also note that the values presented without uncertainties are accurate to all given digits in the caption of Table 2. The contribution referred to as "hadronic effects" is meant to be "hadronic vacuum polarization" and is now corrected and clarified in the text following Table 2. This contribution is nuclear structure independent since it comes from the electron anomalous magnetic moment [see, e.g., arXiv:1208.4194]. However, we have now added a row with the nuclear polarization contributions different for the considered isotopes. Indeed, the uncertainty of the nuclear polarization sets the limit for further improvements of the method presented. Therefore, we have carefully analyzed its uncertainty to the g factor difference and added the corresponding text to Section 3.4.

Table 1: The theory values for the recoils are cut off at 3 digits behind the decimal point, with no uncertainties given. For the comparison between theory and experiment, this is of course sufficient because of the dominant uncertainty from FNS. However, when the argument is turned around (line 310), this appears insufficient. The authors state that in this case (assuming the theory is sufficiently trusted), the value for the charge radius difference can be improved by approximately 1 order of magnitude. Since the previously measured value leads to an uncertainty of 0.011 for the FNS, a 10x improvement would nail down the FNS to 0.001 (assuming FNS uncertainty goes linear with charge radius uncertainty). At that point, the other theory values in Table 1 would be too inaccurate, as they have to be at least +/- 1 digit. While better numbers are found in the supplementary materials, I believe that the main paper must be able to survive standalone. I suggest to add one more digit in Table 1. In this context, it is also apparent that the cancellation of hadronic effects between the isotopes should be at least 10% or better. That should be motivated."

We thank the Referee for this comment. The 4th digit is given now for all contributions in Table 1. Moreover, we added the contribution of the nuclear polarization, which contributes the the next-to-leading uncertainty and determines further improvement in the determination of the rms. Its uncertainty estimation is discussed now in the supplementary material, Section 3.1.

"I have concerns about the 'new' physics section though. Overall, again, the motivation is sound, and I have no issues with the findings in terms of the (TW) curve shown in Figure 3. I also greatly appreciate this approach in the sense that it eliminates this more than awkward reliance on the linearity of the King Plot (in this context, the authors could also cite arXiv:2111.01429v1 (Figueroa et al), which weakens the Yb case from Counts et al.)."

We are pleased to see the Referee's interest in our alternative approach to the well-known King plot analysis. We do agree that "reliance on the linearity of the King Plot" can be "awkward". We have added the relevant reference of Figueroa et al suggested by the Referee.

"However, the section on 'new' physics in the main manuscript is exceedingly short and gives few details. It gives the feeling of a bolt-on section. Important information like what the other curves represent, should in my opinion be addressed in the main body, not just the supplement. If this is an issue of a page limit, I suggest that the description of the experimental method be shortened to create space to work this out in better detail."

We thank the Referee for this suggestion. We have extended and re-written this part of the main text to add more information, especially about prospects for improved bounds in the near future – see our response to your next point below. The caption of Fig. 3 (main text) now names the origin of all the shown bounds coming from other areas of physics.

"Finally: If the reader is assumed to take the H-D 1s-2s and (g-2)n curves seriously, then the manuscript makes no attempt to justify why the current result would be considered competitive. Everywhere in this graph, TW seems to be at least 1.5 orders of magnitude weaker than the best limit (this was already the problem in Counts' paper). In the supplement the authors comment briefly on this fact, but then quickly deflect to discussing the advantages over the King Plot methods, which also don't seem to be competitive. While the statement in the abstract "allows setting constraints" is technically not wrong, this is normally assumed to mean "allows setting new best constraints" in the Beyond-the-Standard-Model community. At the minimum, the result should lend strong support to an existing limit. To me, the TW curve does not seem to do that. However, if the authors could convincingly argue that there could be issues with H-D 1s-2s and/or (g-2)n, or that these limits have some model-dependencies built in that could potentially end up weakening them (say, reliance on a certain proton radius value?), then possibly a case could be made."

The Referee raises a valid point, the bounds derived from the Ne measurements in this work are indeed less competitive than the H-D 1s-2s and (g-2)n bounds. More stringent bounds are projected with the g factor difference approach used in the present work, through combinations of the H-like and Li-like g factors, and combinations of the H-like g factor and the 1s binding energy. These approaches, devised to eliminate nuclear structural effects, largely suppress the dependence on the nuclear radii and their uncertainties. H-D 1s-2s spectroscopy relies on nuclear radii from muonic atom spectroscopy, introducing a possible muonic coupling, and thus complicating the interpretation of possible New Physics phenomena. This is not the case for our projected schemes involving further g factor isotope shifts. This is explained in the revised main text:

"At the present, our method and H-D spectroscopy are both dependent on charge radii determined by muonic spectroscopy [23, 26]. When combining our method with measurements on the g factor of Li-like ions [27] or the ground-state energy isotope shifts of H-like ions, improved competitive bounds could be envisaged [28] while additionally gaining independence from assumptions on NP coupling to muons."

Furthermore, the following was added in the Methods section:

"A similar approach based on a weighted difference of the g factor and ground-state energies of H-like ions should yield even more stringent bounds [6]. Both these weighted-difference-based approaches are insensitive to uncertainties on the nuclear radii, as such, the bounds that they can generate are not only more stringent, but fully independent of any assumptions on NP coupling to muons."

As far as the “(g-2)n” curve is concerned, i.e. the bound derived from a combination of the electron g-2 and neutron scattering experiments: as a comparison of our green curve with the blue (g-2)n curve in Fig. 1 shows, the latter cannot constrain the hypothetical scalar boson coupling constant at boson masses above approx. 50 electron masses. This is due to the lack of neutron scattering data in that regime. The same limitation does not hold for our extracted limits. In this mass regime, the radial extent of the Yukawa potential (i.e. the Compton wavelength of the boson) is still significantly larger than the nuclear radius, and thus our precision atomic spectroscopy can deliver a meaningful limit. Furthermore, since neutron scattering is a completely different domain of physics, our spectroscopic result can become a welcome addition to the armory of New Physics-tests.

List of suggestions:

L3: "the electron's magnetic moment"

Likely to be more intuitive for readers, this has been updated.

L6: only relatively benign nuclear properties come into play, so I'm not sure whether "intricate" is the right word here.

The phrasing has been changed to highlight more clearly that the study of the nuclear effects, which are typically much smaller than QED contributions and their uncertainties, becomes possible in isotopic differences, which is one of the motivations to perform this measurement.

L19: see comments above about what the expectation for "setting constraints" is.

We agree and have changed the phrasing to provide more clarity. The outline and general idea should now become clear: At this point, we present this method foremost as an alternative approach to derive such limits as one of its potential applications.

Table 1: FNS should be written out or defined in the caption or main text before it is used here.

Table 1: I suggest adding one digit to the recoil numbers.

The description has been added to the caption, the additional digits as well as the limiting uncertainty beyond the FNS have been added. See also Table 2 and the more detailed answer to the points given before.

L121-126: Seems to me that this is not a valid sentence, structure-wise.

Corrected, amounts has been changed to amounting

L167: This is the first time that timing is mentioned. Best to say "this takes about 10 min", especially since this is later done repeatedly.

This has been added and clarified each time a duration of the process is mentioned.

Figure 1: "Trap centre" looks not well defined. Maybe it could have a dot assigned to it, just like r_com, with another arrow between them.

This was meant to be shown by the chosen color coding; the blue dot is the trap centre. Added an additional arrow for clarity.

Figure 2: the grey box could use a somewhat lighter grey

Figure 2: the green data point in the lower plot is very hard to spot, make it larger, maybe a filled circle instead of a cross.

Both suggestions have been implemented, thank you for addressing and improving the presentation of the results in this figure.

L316: see above about "setting constraints".

We are confident this to be clarified now in the updated revision.

Figure 3: the caption or the main text should refer to the other experiments, at least the ones that actually seem to set stronger limits.

References have been added to the caption, with more detailed information in the Methods section.

References:

L440: this does not look like an arXiv citation.

Correct, the reference has been updated accordingly.

Supplement: L765: is the correct citation for the $(g-2)_n$ work really reference 19, Solaro, which is on the Ca+ King Plot work, or is that a mistake?

This was a mistake; the original measurements are now cited. While the limits also appear in Solaros work, they are first explicitly derived in the work of Berengut et al., Ref. [31] in the updated manuscript.

We thank the Referee for the careful reading of our work and the suggestions that have helped to improve the addressed parts of the manuscript.

Referee #3:

However, I do find a slightly different number for the systematic uncertainty in the final result if I follow the numbers in the paper:

Eqn (29) states a relative uncertainty in the Lamor frequency difference of 6×10^{-13} (incidentally this differs from the definition on line 601). If I multiply 6×10^{-13} by 112 GHz and divide by the Lamor frequency difference of 758.75 Hz, I get 8.8×10^{-5} for the fractional uncertainty in the g factor difference, corresponding to 0.00119×10^{-9} rather than 0.00099 for the systematic uncertainty. I guess this is could be a rounding error?

Equation 29 in the methods is used to calculate the expected relative systematic **shift** of the difference frequency: $\frac{\Delta(\Delta\nu_L)}{\nu_L} \approx 6 \times 10^{-13}$, and needs to be corrected for in the final result. The **uncertainty** of this correction is given below in line 601 as 5×10^{-13} .

If the systematic uncertainty is used as 5×10^{-13} , the given uncertainty of $13.475\ 24(99) \times 10^{-9}$ is correct.

One thing that was not obvious to me at first, and that could be emphasized in the paper is the fact that this technique significantly reduces the demand on the precision of the cyclotron frequency measurement and that only the cyclotron frequency of one ion needs to be measured at all to obtain the difference in g. This is stated somewhat passively in the first sentence of section 3.3.

I say this because on lines 148-155, the authors emphasize how much a direct measurement of g is limited by the cyclotron frequency measurement, and also the masses of the ion and the electron. Perhaps providing the formula for Δg (in the methods section) would help make this clearer.

The formula for Δg has been added in section 2.3, together with an explanation of this benefit.

Line 41-45: "However, studying these effects...(see Methods 2)".

Do the authors mean Methods 3.4, g factor calculation?

Yes, this should have been given as "Methods, Table 2" and has been updated to be consistent with the reference in the caption of Table 1 now.

Line 55-58: "For the calculated difference $\Delta g = \dots$ the absolute uncertainty is decreased by two orders of magnitude compared to the absolute values..."

Perhaps I am not interpreting this statement correctly or missing something. The absolute uncertainty in Δg is 0.011×10^{-9} . What do the authors mean by "compared to the absolute values"? I took this to mean the uncertainty in the absolute g factor values, which is 1.17×10^{-10} . The ratio of these two numbers is one order of magnitude, not two.

This interpretation is correct, the reduction of the uncertainty is indeed only 1 order of magnitude and does not affect the underlying statement. However, the section has been updated to clarify that the QED Recoil contribution is meant to be tested in the difference, where other common effects do not have to be considered.

Table 1: The abbreviation FNS is used here. It is defined in the Methods section, but not here or in the main text.

The definition has been added to the caption of Table 1.

Line 121 – 126: “Additionally, the presence of ... with the Larmor frequency $\nu_L = \dots$ [that] amounts to about 112 GHz, ...”

This has been corrected: Additionally, the presence of... with the Larmor frequency $\nu_L = \dots$ amounting to about 112GHz..

Line 175 – 179: “The combined motion, ... around a common guiding center on a radius r_{com} and a rotation of this guiding center around the trap center.” This sentence as written is not completely clear.

Suggestion: “The combined motion, ... around a common guiding center, that rotates around the trap center on a radius r_{com} .”

The structure of the sentence has been changed to clarify the superposition of separation mode and the rotation of the guiding centre on the common radius with respect to the geometrical trap centre.

Line 187 – 191: Subsequently we are able to convert common mode into separation mode radius...” The language used here is a little loose. As I understand it, it is the classical action that is transferred between normal modes of an ion in the Penning trap. Is this also the case for the separation and common mode? Is there a one-to-one correspondence between the radius of common and separation modes after this conversion?

Indeed, the coupling process ideally conserves canonical angular momentum or in this case equivalently the classical action, with the subtlety that the separation frequency depends on the separation distance. However, in our experimental condition, we can assume the frequencies to stay largely constant and even almost identical. Consequently, we expect an (almost) 1:1 correspondence of the radii as well.

Due to the current imperfect experimental implementation this cannot be ensured here and the transfer is likely not completely adiabatic. Furthermore, we can only give upper limits on the final common mode and therefore also not give a quantitative description for the transfer at this point.

We have updated the corresponding section to provide more clarity as:

“Subsequently we are able to transfer canonical angular momentum, or effectively mode radius, from common- to separation mode, as well as directly...”

In combination with the methods section and the descriptions in the pioneering work of Ref. [20], we believe this to be clarified.

Line 193 – 196: “We use these tools to prepare the ions with ... $d_{sep} = 400 \mu\text{m}$ and a comparably small common mode radius r_{com} .”

Is the common mode radius not made to be approximately equal to 0?

This is correct, the common mode is decreased as far as possible. However, the experimental resolution of the common mode radius is limited and at this point, we could only verify the common mode radius to be smaller than $100 \mu\text{m}$ at all times at the end of this conversion. We discuss this in

more details in the methods “Mixing and preparing the coupled state”. Due to the same reason, we also cannot guarantee or measure a “perfect” (or one-to-one) conversion of common to separation mode.

Line 586: “...larger than for $^{22}\text{Ne}9+[\cdot]$ ”

Corrected.

Line 602: “This [has] been confirmed...”

Corrected.

Line 628: “inferred, without ... common mode [however].” Remove however.

Corrected, and clarified: at this point, we only gain information about the separation distance.

Line 670: “The strong magnetic bottle...gives rise to a force [that is] dependent on...”

Corrected.

Organization of Methods section. The structure of the Methods section is not entirely intuitive, e.g. Section 2 is titled “Calculation of the initial phase”, but should really be the first sub-section of section 2, because subsection 2.1 is “Combined systematic shifts...”, which isn’t a subsection under the main heading of section 2. The same applies to subsection 2.2.

The Methods section has been updated and restructured to follow a more logical approach, meant to guide the reader through additional information about how the measurement was performed.

Section 3 “Ion manipulation” is appropriate for subsections 3.1, 3.2 and 3.3, but subsection 3.4 “g factor calculation” doesn’t fit under this section heading. The same is true for 3.5.

Section headings and structure has been updated and improved, thank your for the comment. The method section should now be easier accessible.

We again thank all Referees for their suggestions, which further helped to improve the manuscript.

On behalf of all co-authors,

Tim Sailer